# Successor Clusters: A Behavior Basis for Unsupervised Zero-Shot Reinforcement Learning

**Louis Bagot**                                                    *louis.bagot.nao@gmail.com*
*IDLab - Department of Computer Science, University of Antwerp - imec*

**Lucas N. Alegre**                                                    *lnalegre@inf.ufrgs.br*
*Institute of Informatics - Universidade Federal do Rio Grande do Sul*

**Steven Latré**                                                    *steven.latre@uantwerpen.be*
*IDLab - Department of Computer Science, University of Antwerp - imec*

**Kevin Mets**                                                    *kevin.mets@uantwerpen.be*
*IDLab - Department of Electronics and Information and Communication Technology, Faculty of Applied Engineering, University of Antwerp - imec*

**Bruno Castro da Silva**                                                    *bsilva@cs.umass.edu*
*University of Massachusetts*

**Reviewed on OpenReview:** *https://openreview.net/forum?id=UB22Tt3sfF*

## Abstract

In this work, we introduce *Successor Clusters* (SCs), a novel method for tackling unsupervised zero-shot reinforcement learning (RL) problems. The goal in this setting is to directly identify policies capable of optimizing *any* given reward functions without requiring further learning after an initial reward-free training phase. Existing state-of-the-art techniques leverage Successor Features (SFs)—functions capable of characterizing a policy's expected discounted sum of a set of $d$ reward features. Importantly, however, the performance of existing techniques depends critically on how well the reward features enable arbitrary reward functions of interest to be linearly approximated. We introduce a novel and principled approach for constructing reward features and prove that they allow for *any* Lipschitz reward functions to be approximated arbitrarily well. Furthermore, we mathematically derive upper bounds on the corresponding approximation errors. Our method constructs features by clustering the state space via a novel distance metric quantifying the minimal expected number of timesteps needed to transition between any state pairs. Building on these theoretical contributions, we introduce *Successor Clusters* (SCs), a variant of the successor features framework capable of predicting the time spent by a policy in different regions of the state space. We demonstrate that, after a pre-training phase, our method can approximate and maximize *any* new reward functions in a zero-shot manner. Importantly, we also formally show that as the number and quality of clusters increase, the set of policies induced by Successor Clusters converges to a set containing the optimal policy for *any* new task. Moreover, we show that our technique naturally produces *interpretable* features, enabling applications such as visualizing the sequence of state regions an agent is likely to visit while solving a task. Finally, we empirically demonstrate that our method outperforms state-of-the-art SF-based competitors in challenging continuous control benchmarks, achieving superior zero-shot performance and lower reward approximation error.

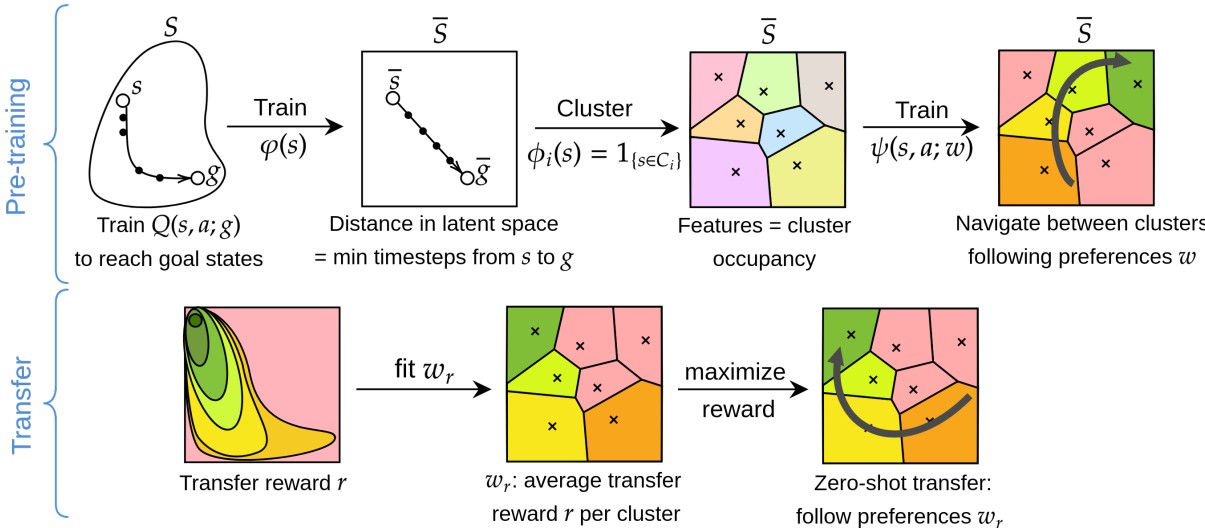

Figure 1: The *Successor Cluster* framework. **Pre-training:** The agent builds a latent space, $\varphi(\mathcal{S})$, that encodes a distance function based on the minimum number of time steps required to reach any state via goal-reaching RL. This is achieved by learning a goal-conditioned action-value function, $Q(s, a; g)$. The induced distance function is used to identify a set of clusters of the state space, $\{C_i\}_{i=1}^{d}$, that are then used as features, $\phi_i(s) = \mathbb{1}_{C_i}(s)$. The zero-shot agent learns *Successor Clusters*, $\psi(s, a; \mathbf{w})$, which model the expected discounted time spent visiting each cluster $C_i$ while solving a task with reward function $r_{\mathbf{w}} = \phi \cdot \mathbf{w}$. **Transfer:** Given any reward function $r$, the agent approximates it by fitting the reward weights: $r \approx \phi \cdot \mathbf{w}_r$. Finally, the agent can follow the policy that maximizes the cluster preferences $\mathbf{w}_r$ in a zero-shot manner.

# 1 Introduction

In Reinforcement Learning (Sutton & Barto, 2018, RL), an *agent* interacts with an *environment*, defined as a Markov decision process (MDP), performing *actions* to maximize *rewards* in a sequential task, based on its current *state*. However, for a given environment, we can often define a wide range of *tasks* via different reward functions. For instance, a single robot arm can perform a very vast array of tasks, which might all be relevant in certain contexts. Classic RL algorithms are not equipped to deal with this, as they essentially overfit to a single reward function and would need re-training from scratch to learn to solve another task. The field of *unsupervised zero-shot* RL (Touati et al., 2023) aims to train a general agent, in a reward-free pre-training stage, that is able to instantly adapt—without any further training—to any new reward function in a transfer stage.

In theory, model-based RL alone can achieve zero-shot RL when a highly accurate model and unlimited deep decision-time planning budget are accessible. However, this is almost always impractical in real-world tasks. Model-free RL methods have been extended to perform zero-shot RL, where the general idea is to condition the policy on a task vector, $\mathbf{w} \in \mathbb{R}^d$, that summarizes a task or policy, and train this policy to solve as many tasks as possible. This shifts the problem to that of finding a representative task vector encoding under a dimension budget $d$. Methods based on the Successor Features (SFs) framework (Barreto et al., 2017b; 2018) linearly approximate the reward function from *reward features* $\boldsymbol{\phi}(s) \in \mathbb{R}^d$, $r(s) \approx \boldsymbol{\phi}(s) \cdot \mathbf{w}$. The task vector $\mathbf{w}$ then defines the contribution of each feature in the value of the reward, and the task-conditioned policy maximizes the features according to their weights. However, the zero-shot performance of such policies is known to depend on how well one can approximate the reward function of interest as a linear function of the reward features. This opens the problem of how to find a good reward feature representation $\boldsymbol{\phi}$, so that the set of policies induced by $\boldsymbol{\phi}$ is capable of solving any task in a zero-shot manner. Current best-performing zero-shot RL methods often rely on the graph Laplacian, theoretically providing a Fourier-like linear basis for the MDP (Barto & Mahadevan, 2003; Machado et al., 2018). These methods have recently been bench-marked in an extensive study by Touati et al. (2023).

In this work, we introduce a new method for identifying reward features $\phi$ for SF-based methods constructed by clustering the state space of the MDP. Based on this choice of reward features, we introduce Successor Clusters (SCs), a variant of SFs that predicts the expected time a policy spent visiting each state cluster. We use these to generate zero-shot policies. We show an overview of our method in Figure 1. SCs have important theoretical guarantees (as we list below), and provide state-of-the-art zero-shot performance when compared to existing reward feature construction techniques used in SF-based methods. In summary, the contributions of this work are the following:

- We propose a novel mathematical formulation to the problem of learning reward features in the SFs framework. We show that its solution is bounded by the solution to the problem of clustering the state space of the MDP. This results in reward features defined by state cluster occupancy. Importantly, these features allow for high-quality reconstruction of any Lipschitz reward functions.

- Based on this theoretical result, we introduce Successor Clusters, a novel type of SFs that predict future discounted cluster occupancies of policies. Additionally, we introduce a practical algorithm for unsupervised zero-shot RL based on SCs. We demonstrate that as the number and quality of clusters improve, the set of policies induced by our features converges to a set that contains an optimal policy for any given task.

- To identify state clusters, we introduce a novel notion of state distance based on *temporal proximity*, under which states in a same cluster are closer in terms of the minimal number of time steps necessary to reach each other. We introduce a practical algorithm to learn an associated latent metric space that encodes this distance function. We qualitatively show that this distance metric, latent space, and clustering can be humanly interpreted.

- Compared to existing methods, Successor Clusters (SCs) are highly interpretable and allow for intuitive visualization of the regions of the state space an agent is likely to visit while solving a given task. This stands in stark contrast to the often opaque, black-box nature of many competing approaches. The interpretability of SCs enables us to characterize their behavior in zero-shot reinforcement learning settings—in particular, SCs predict the *expected time spent in each cluster* from a given state, conditioned on a reward function. We also demonstrate that this property makes it possible to infer reward functions based on cluster visitation patterns or trajectories induced by a policy. We believe this form of interpretability can positively impact research and debugging, while also improving transparency and trust in deployed systems.

- We empirically show that SCs outperform state-of-the-art SF-based methods in a challenging unsupervised RL continuous control benchmark in terms of zero-shot performance. Additionally, we quantitatively show that our method leads to better reward approximation performance compared to a state-of-the-art SF-based zero-shot transfer technique.

This work is organized as follows. In Section 2, we summarize the theoretical background necessary to introduce our methods. In Section 3, we introduce our theoretical results to the problem of learning reward features $\phi$, leading to the definition of SCs. We propose a practical learning algorithm for unsupervised zero-shot RL based on SCs in Section 4. We present quantitative and qualitative experimental evaluation results in Section 5. Finally, in Section 6, we discuss how our work fits within the zero-shot research landscape and examine its limitations and potential improvements in Section 7.

## 2 Background

**Reinforcement Learning.** In RL (Sutton & Barto, 2018), the agent-environment interactions are generally formalized via a *Markov Decision Process* (MDP), defined as a tuple $M \triangleq (\mathcal{S}, \mathcal{A}, r, p, \mu, \gamma)$ whose elements respectively represent the state space, action space, reward function, state transition function, initial state distribution, and discount factor. The transition function $p(s' \mid s, a)$ dictates the next state distribution, while the *reward function* $r(s, a)$ dictates the next reward. We denote by $S_t$, $A_t$, and $R_t$ the random variables of the agent's state, action, and reward, respectively, at time step $t$. The goal of an agent

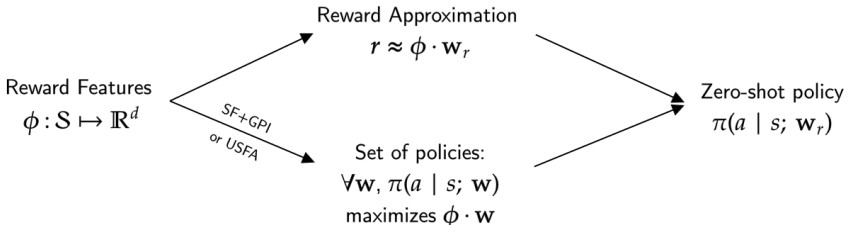

Figure 2: The central role of the reward features $\phi$ in SF-based zero-shot RL. The reward features $\phi : \mathcal{S} \mapsto \mathbb{R}^d$ are used to train a set of policies $\{\pi\,(a \mid s; \mathbf{w})\}_{\mathbf{w} \in \mathcal{W} \subset \mathbb{R}^d}$ that maximize rewards defined as linear combinations $r_{\mathbf{w}} = \phi \cdot \mathbf{w}$—different choices of reward features $\phi$ *induce* different sets of policies. During transfer, they are used to linearly approximate the reward function $r \approx \phi \cdot \mathbf{w}_r$, leading to the zero-shot policy $\pi\,(a \mid s; \mathbf{w}_r)$. Better reward approximation leads to better zero-shot policies.

is to learn a policy $\pi : \mathcal{S} \mapsto \mathcal{A}$ that maximizes the expected value of the return $G_t \triangleq \sum_{k=0}^{\infty} \gamma^k R_{t+k+1}$. The action-value function of a policy $\pi$ is defined as $q^\pi(s, a) \triangleq \mathbb{E}_\pi[G_t \mid S_t = s, A_t = a]$, where $\mathbb{E}_\pi[\cdot]$ denotes expected value with respect to the trajectories induced by $\pi$ and $p$.

**Unsupervised Zero-shot Transfer in RL.** In this work, we focus on a specific type of transfer in RL in which the dynamics are fixed but the agent has to adapt to any task defined by different reward functions in a *zero-shot* manner. We assume a first stage of training where the agent interacts with a Markov Control Process (MCP), i.e., an MDP *without a specific reward function*, and learns some form of knowledge basis in an unsupervised manner. The second stage consists of presenting a reward function to the agent, either via the analytic function $r\,(s, a)$, or via a set of samples $\{r(s_i, a_i)\}_{i=1}^n$. The zero-shot RL objective is to derive a policy that maximizes the return of this reward function without any additional training or interactions with the environment. This problem is closely related to Unsupervised Skill Discovery (USD) (Eysenbach et al., 2019a; Hansen et al., 2020), which aims to build a task-agnostic skill set for use in any downstream task. However, USD typically trains a meta-agent to use the skills during transfer.

**Successor Features.** Let $\phi : \mathcal{S} \times \mathcal{A} \mapsto \mathbb{R}^d$ be a *reward feature* function (i.e., a vector of $d$ reward-like functions, also called "cumulants"). Successor Features (SFs) (Barreto et al., 2017b) allow us to decompose the action-value function $q^\pi$, such that it is possible to instantly compute the $q$-values of policies given *any* reward function linearly-expressible over $\phi$ with weights $\mathbf{w} \in \mathbb{R}^d$:

$$r_{\mathbf{w}}\,(s, a) = \phi\,(s, a) \cdot \mathbf{w} \implies q_{\mathbf{w}}^\pi\,(s, a) = \psi^\pi\,(s, a) \cdot \mathbf{w}, \tag{1}$$

where $\psi^\pi\,(s, a) = \mathbb{E}_\pi\left[\sum_{k=0}^\infty \gamma^k \phi\,(S_{t+k}, A_{t+k}) \mid S_t = s, A_t = a\right]$ are the SFs of policy $\pi$, and $q_{\mathbf{w}}^\pi(s, a)$ is its action-value function w.r.t. task $\mathbf{w}$. This result extends to the approximate scenario $r \approx \phi \cdot \mathbf{w}$, where the performance w.r.t $r$ of an agent maximizing $\phi \cdot \mathbf{w}$ improves with the quality of the approximation (Barreto et al., 2018). This decomposition is of critical importance because it allows us to perform generalized policy evaluation (GPE) (Barreto et al., 2020) efficiently, i.e., to evaluate a policy $\pi$ over a *set* of tasks: $q_{\mathbf{w}'}^\pi(s, a) = \psi^\pi(s, a) \cdot \mathbf{w}', \forall \mathbf{w}' \in \mathbb{R}^d$. Similarly to an action-value function, $\psi^\pi$ follows a Bellman equation and can be learned via any temporal-difference (TD) algorithm. Notice that in order to exploit the decomposition property provided by SFs, an agent needs to learn the SFs of a set of policies, so that it can use the most effective ones for any given $\mathbf{w} \in \mathbb{R}^d$ during transfer.

**Universal SF Approximators.** Universal SF Approximators (USFAs) (Borsa et al., 2019) were proposed to deal with the fact that learning independent SFs, $\psi^\pi\,(s, a)$, for each agent's policy $\pi$ may be computationally prohibitive. USFAs draw inspiration from goal-conditioned neural network architectures, such as Universal Value Functions Approximators (UVFAs) (Schaul et al., 2015), and employ a *single* approximator $\psi\,(s, a; \mathbf{w})$ which is conditioned on task vectors $\mathbf{w}$ encoding the reward function. When using USFAs, the greedy policy $\pi_{\mathbf{w}}(s) \in \arg\max_{a \in \mathcal{A}} \psi\,(s, a; \mathbf{w}) \cdot \mathbf{w}$ approximates the optimal policy for $r_{\mathbf{w}}$, i.e.,

$q_{\mathbf{w}}^*(s,a) \approx \boldsymbol{\psi}(s,a;\mathbf{w}) \cdot \mathbf{w}$. Crucially, USFAs aim to generalize over the space of weight vectors $\mathbf{w} \in \mathbb{R}^d$ in order to represent multiple optimal policies.

## 3 Successor Clusters as a Representation for Zero-shot Transfer

Our goal is to learn a good reward representation, $\boldsymbol{\phi} : \mathcal{S} \mapsto \mathbb{R}^d$, that can be used with SF-based methods for unsupervised zero-shot transfer.[1] Intuitively, we aim to learn reward features that induce a good behavior basis for solving new tasks without additional training. Given reward features, $\boldsymbol{\phi}$, such behavior basis is induced by learning policies optimized to solve tasks defined by linear rewards $r_{\mathbf{w}} = \boldsymbol{\phi} \cdot \mathbf{w}$, for all $\mathbf{w} \in \mathcal{W}$, where $\mathcal{W} \subset \mathbb{R}^d$ denotes the space of linear reward weight vectors.

Critically, the performance of a zero-shot SF-based agent is mathematically tied to how well we can approximate rewards as a linear function of the reward features. Let $r$ be an arbitrary reward function, and let $\mathbf{w}$ be the weight vector that best approximates $r$ given features $\boldsymbol{\phi}$: $r \approx r_{\mathbf{w}} = \boldsymbol{\phi} \cdot \mathbf{w}$. Then, assuming access to a policy $\pi_{\mathbf{w}}$ optimal w.r.t. $r_{\mathbf{w}} = \boldsymbol{\phi} \cdot \mathbf{w}$, Barreto et al. (2018) showed that:

$$\|q_r^* - q_r^{\pi_{\mathbf{w}}}\|_\infty \leq \|r - \boldsymbol{\phi} \cdot \mathbf{w}\|_\infty / (1 - \gamma), \tag{2}$$

where $q_r^*$ is the optimal action-value function for $r$ evaluated on $r$, and $q_r^{\pi_{\mathbf{w}}}$ is the value function of $\pi_{\mathbf{w}}$ also evaluated on $r$. In other words, the optimality gap of $\pi_{\mathbf{w}}$ (i.e., the difference between $q^*$ and $q^{\pi_{\mathbf{w}}}$ when evaluated on $r$) depends on how well the features $\boldsymbol{\phi}$ can be used to approximate $r$. **Therefore, in this work, our primary objective is to construct the best possible reward features**, $\boldsymbol{\phi}(s) \in \mathbb{R}^d$, to be used with SF-based algorithms. We start by introducing a novel mathematical objective for the problem of identifying reward features in Section 3.1. Next, we connect this objective to a problem of clustering the state space of the MDP in Section 3.2), and finally introduce *Successor Clusters* (SCs), the main contribution of this work, in Section 3.4.

### 3.1 A Formal Objective for Learning Reward Features

The problem of reward approximation is central to constructing features for successor features (SFs), since optimality is bounded by the quality of the approximation $r \approx \boldsymbol{\phi} \cdot \mathbf{w}$ for any given reward function $r$, as shown in the previous section. Let $\Gamma \triangleq \{r \mid r : \mathcal{S} \mapsto [r_{\min}, r_{\max}]\}$ denote the space of all reward functions bounded between $r_{\min}$ and $r_{\max}$, and let $\Phi^d \triangleq \{\boldsymbol{\phi} \mid \boldsymbol{\phi} : \mathcal{S} \mapsto \mathbb{R}^d\}$ denote the space of possible reward feature functions of dimension $d$. Intuitively, our goal (formally defined in Equation (3)) is to identify a feature representation $\boldsymbol{\phi} \in \Phi^d$ that enables broad and accurate linear approximation of reward functions in $\Gamma$. That is, we seek features that span the "widest" possible range of reward functions—ensuring that, across many possible rewards, $\boldsymbol{\phi}$ supports accurate linear approximations. Formally, this objective captures the goal of minimizing the expected mean squared error between each reward function in $\Gamma$ and its best linear approximation induced by $\boldsymbol{\phi}$. The error is computed over the state space $S$, and the expectation is taken with respect to a distribution $\mathcal{P}$ over reward functions likely to arise during transfer.

We call this objective **reward coverage maximization** and formalize it as follows:[2]

$$\min_{\boldsymbol{\phi} \in \Phi^d} \mathbb{E}_{r \sim \mathcal{P}(\Gamma)} \left[ \min_{\mathbf{w} \in \mathbb{R}^d} \sum_{s \in S} (r(s) - \boldsymbol{\phi}(s) \cdot \mathbf{w})^2 \right]. \tag{3}$$

Note that, in general, we do not have prior knowledge of the distribution over relevant—or likely—reward functions the agent may encounter. When the state space $S$ is finite, a reward function can be represented as a vector $r \in \mathbb{R}^{|S|}$, and $\mathcal{P}$ can be taken to be any distribution over $\mathbb{R}^{|S|}$. A natural and mathematically grounded choice—especially when little is known a priori about the structure of reward functions—is to adopt a *maximum entropy distribution*, which corresponds to the distribution that makes the fewest assumptions beyond basic constraints. For instance, if we only assume that the expected squared norm $\mathbb{E}[\|r\|^2]$ is bounded,

---

[1]We will restrict our study to state-based rewards $r(s') = r(s, a, s')$ for simplicity. We note, however, that all definitions can be easily extended to employ general reward features $\boldsymbol{\phi}(s, a, s')$ instead.

[2]Without loss of generality, in continuous spaces, the sum over the state space can be replaced by an integral.

then the maximum entropy distribution is a multivariate Gaussian with zero mean and isotropic covariance: $r \sim \mathcal{N}(0, \sigma^2 I)$. Alternatively, if we assume that reward values are bounded (e.g., $r(s) \in [r_{\min}, r_{\max}]$ for all $s$), then the maximum entropy distribution becomes the uniform distribution over the hypercube $[r_{\min}, r_{\max}]^{|S|}$. Both cases are useful in practice and provide principled ways to define expectations over reward functions while maintaining minimal prior assumptions.[3]

To the best of our knowledge, this is the first formal definition of an objective for learning features with the goal of approximating as many reward functions as possible in the SFs literature. The Fourier basis is an intuitive approach to this problem—if we can find an equivalent for MDPs. Proto-value functions (PVFs) (Barto & Mahadevan, 2003) propose a solution to a very similar problem by finding a basis over value functions through an eigenvalue decomposition of the graph Laplacian. PVFs provide Fourier-like basis functions that are smooth along the MDP geodesic directions. Eigenfunctions are currently the main approach for learning features in SF-based methods. However, estimating them in high-dimensional spaces remains challenging. This has led to significant recent research focused on improving Laplacian computation and decomposition techniques (Wu et al., 2019; Wang et al., 2021; Gomez et al., 2024). Critically, for any choice of feature basis, the dimension $d$ is a significant constraint: current state-of-the-art methods struggle to learn them when $d > 100$ (Touati et al., 2023). Therefore, in practice, we have a very limited budget on the number of features $d$.

The most important element for deciding a basis $\phi$ is the family of rewards $\Gamma$ considered. In the extreme case, a valid reward $r \in \Gamma$ has arbitrarily random values at every single state. Arguably, this type of reward does not correspond to any practical task, and thus, some constraints on $\Gamma$ are acceptable. In other RL problem settings, works have proposed either piecewise or Lipschitz constraints in the reward functions to ensure that tasks make practical sense (Khetarpal et al., 2022). To be best of our knowledge, there is no similar consensus on a subset of $\Gamma$ that is considered relevant and practical, but we think this discussion would be justified. In this paper, we will be focusing on $K$-Lipschitz reward functions: $\Gamma_{K\text{-}Lip} = \left\{ r : \mathcal{S} \mapsto \mathbb{R} \mid \forall s, s' \in \mathcal{S}, (r(s) - r(s'))^2 \leqslant K\delta(s, s') \right\}$. This assumption guarantees that rewards do not fluctuate too drastically w.r.t. some state distance metric $\delta$.

## 3.2 Reward Features via Clustering of the State Space

In this section, we propose a novel approach to tackle the reward coverage maximization problem (Equation (3)) via a partitioning of the state space $\mathcal{S}$ of the MDP. The proofs of the theorems introduced in this paper can be found in Appendix A.1. We start by introducing a novel theoretical result that shows that, for $K$-Lipschitz reward functions, we can approximate a solution to the reward coverage maximization problem by solving a clustering problem over the state space of the MDP:

**Theorem 1.** *Let $\mathcal{C} = \{C_i\}_{i=1}^d$ be a partitioning of the state space $\mathcal{S}$ and let $\Gamma_{K\text{-}Lip}$ be the space of $K$-Lipschitz reward functions for some distance metric $\delta$ over $\mathcal{S}$. Let $\phi(s) = [\mathbb{1}_{C_1}(s), \cdots, \mathbb{1}_{C_d}(s)]^\top$ be the feature vector indicating subset occupancy: $\mathbb{1}_{C_i}(s) = 1$ if $s \in C_i$ and $0$ otherwise. Then, reward coverage maximization is bounded as follows:*

$$\mathbb{E}_{r \sim \mathcal{P}(\Gamma_{K\text{-}Lip})} \left[ \min_{\mathbf{w} \in \mathbb{R}^d} \sum_{s \in \mathcal{S}} (r(s) - \phi(s) \cdot \mathbf{w})^2 \right] \leq K \sum_{i=1}^d \frac{1}{|C_i|} \sum_{x,y \in C_i} \delta(x, y). \tag{4}$$

Notice that the upper bound above depends on the choice of how to partition the state space, $\{C_i\}_{i=1}^d$. Minimizing this upper bound can be achieved by solving a *clustering problem* whose goal is to identify clusters (partitions of $\mathcal{S}$) that minimize the intra-cluster distances. Such families of problems can be efficiently addressed via standard clustering algorithms, e.g., K-medoids (Kaufman, 1990) or K-Means (Lloyd, 1982). We note that we have a strictly improving bound as $d$ increases: adding a cluster will always result in a

---

[3]In more general settings, where the state space is continuous or infinite, $\mathcal{P}$ can be defined as a distribution over functions. A common choice is to use a Gaussian process (GP) prior, which generalizes the Gaussian case to infinite-dimensional function spaces. If the reward function is assumed to lie in a bounded interval, as above, then a simple and flexible prior is a GP with a constant mean function (e.g., $(r_{\min} + r_{\max})/2$) and a stationary kernel such as the squared exponential (RBF) kernel with large length-scale. This setup encourages smoothness while avoiding strong structural biases.

strictly better approximation, e.g., by splitting an existing cluster in two. An important intuition on this choice of features is that, at any given time-step $t$, $\phi(S_t)$ indicates which cluster the agent is currently in: $\phi$ is the *cluster occupancy* vector. Additionally, when using this representation, the optimal $\mathbf{w}$ corresponds to a discretization of the reward function over the clusters: $w_i$ is the average reward in cluster $C_i$ (see proof in A.1). Since we assume Lipschitz reward functions, states that are identified in the same cluster by some clustering algorithm would likely have similar values of rewards. The restriction of the objective to the set of Lipschitz rewards is critical to this analysis. In Section 7.1, we further discuss this assumption and its impact on the performance of the zero-shot policies induced by the reward features introduced in this section.

### 3.3 On Distance Functions for Clustering

In order to identify clusters $C_1, \ldots, C_d$ that induce features $\phi(s) = [\mathbb{1}_{C_1}(s), \cdots, \mathbb{1}_{C_d}(s)]^\top$, to minimize the upper-bound in Equation (4), we need to define a distance function $\delta(s, s')$ over states $s, s' \in \mathcal{S}$. This raises the question: which state distance function $\delta(s, s')$ to use when identifying clusters $\mathcal{C} = \{C_i\}_{i=1}^d$ via a clustering algorithm? In theory, the best distance function to use depends on the class of reward functions, $\Gamma$, being approximated. Distance functions that rely directly on the vector representation of a state are unlikely to be helpful in practice since they might incorporate implicit biases and are not linked to the topological properties of the state space (e.g., pixel representations, or $(x, y)$-coordinates in grid-worlds with obstacles).

We draw inspiration from stochastic shortest paths (Bertsekas & Tsitsiklis, 1991), and propose to use a notion of distance based on the minimal number of time-steps it takes to get from one state to another. This is independent of the original state representation and gives a direct notion of connectedness within the MDP graph. Spectral analysis of the *graph Laplacian* (Machado et al., 2018; Wu et al., 2019) and *average commute time* (Venkattaramanujam et al., 2019) compute such metrics, but provide the expected time-steps under a *fixed policy* (e.g., a uniformly random or exploration policy). Instead, we aim to use the *shortest* commute time under an optimal goal-reaching policy. This better represents distances under efficient navigation policies, which we expect will be more useful for zero-shot transfer.

Goal-conditioned RL allow us to compute such a distance for any two states. Assuming any two states are connected (i.e., that for any two states, there exists a *proper stationary policy*), finding the shortest stochastic path between states $s$ and $g$ is equivalent to finding the optimal policy from $s$ under constant reward $-1$ and terminal state $g$ (Bertsekas & Tsitsiklis, 1991). The negative value function then exactly corresponds to the distance function of interest: the minimal expected number of time-steps to go from $s$ to $g$ is $-\max_\pi v_\pi(s; g)$, the optimal value function conditioned on goal $g$. Based on this idea, we introduce the *average minimal path* distance, a symmetric state distance function defined as

$$d_\star(s_1, s_2) \triangleq -\frac{1}{2} \max_\pi v_\pi(s_1; s_2) - \frac{1}{2} \max_\pi v_\pi(s_2; s_1). \tag{5}$$

where $v_\pi(s_1; s_2)$ is the value function of $\pi$ for state $s_1$ with goal (terminal state) $s_2$. Using the distance $d_\star$ defined above to minimize the upper-bound in Equation (4) has implications on the reward space $\Gamma$ and the resulting clustering. A $K$-Lipschitz reward function under $d_\star$ implies that the reward difference between any two connected states will always be lower than $K$. Identifying clusters that minimize Equation (4) under $d_\star$ establishes a notion of temporal proximity: states within the same cluster are reachable in a short number of time steps. In Section 4.2 we will discuss how to approximate $d_\star$ in practice.

### 3.4 Successor Clusters: Predicting Future Cluster Occupancy

Based on the theoretical motivation for using reward features $\phi(s)$ defined via partitioning the state space of the MDP, introduced in Theorem 1, we are now ready to introduce *Successor Clusters*, the central contribution of this work:

Given a partitioning $\mathcal{C} = \{C_i\}_{i=1}^d$ of the state space $\mathcal{S}$, the *Successor Clusters* (SCs) of a policy $\pi$ for a state-action pair $(s, a)$ is defined as:

$$\boldsymbol{\psi}^\pi(s, a) = \mathbb{E}_\pi\left[\sum_{k=0}^\infty \gamma^k \boldsymbol{\phi}(S_{t+k+1}) \mid S_t = s, A_t = a\right], \quad \text{where} \quad \boldsymbol{\phi}(s) = [\mathbb{1}_{C_1}(s), \cdots, \mathbb{1}_{C_d}(s)]^\top. \quad (6)$$

It follows from our choice of $\boldsymbol{\phi}$ that **each $i$-th component of the SCs vector $\boldsymbol{\psi}^\pi(s, a) \in \mathbb{R}^d$, i.e., $\boldsymbol{\psi}^\pi(s, a) = [\psi_1^\pi(s, a), \cdots, \psi_d^\pi(s, a)]^\top$, represents the expected discounted time spent visiting each cluster $C_i$ when following policy $\pi$.**

Notice that SCs are a type of SF whose values are naturally more interpretable. Intuitively, since $\boldsymbol{\psi}^\pi$ characterizes the discounted number of steps spent in each cluster $C_i$, it also encodes the expected amount of time $\pi$ spends in each region of the state space. In Figure 3, we depict an example of the SCs of a policy to illustrate this idea. SCs therefore enable us to easily interpret the outcome of a policy without needing to run it: while it does not detail the order a policy visits each cluster, it summarizes the policy in terms of spatial occupancy. This interpretability can be an advantage when employing SCs. When learning an optimal policy, it can help us understand its trajectory and inform us on the geometry of the domain and task. When learning a function approximator, it can be easier to detect, identify and correct errors, since we have a more intuitive understanding of what outputs to expect.

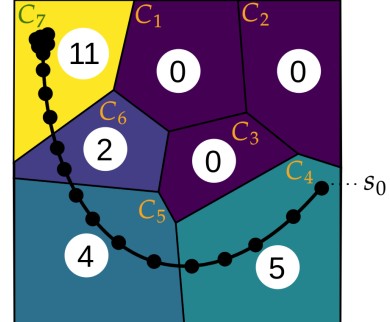

Figure 3: The Successor Clusters associated with the policy in black are $\boldsymbol{\psi}^\pi(s_0, \pi(s_0)) = [0, 0, 0, 5, 4, 2, 11]$. We assume $\gamma = 1$ and consider a finite-horizon problem in which each episode terminates after a fixed, predetermined number of steps.

We also note that $\sum_{k=1}^d \psi_k^\pi(s, a) = \frac{1}{1-\gamma}$,[4] for any given state-action pair $(s, a)$. Hence, we can see SCs as a type of *occupancy measure* (Laroche & Tachet Des Combes, 2023) defined over regions of the state space instead of single states. Notice that the *Successor Representation* (Dayan, 1993, SR) of a policy $\pi$, $\psi^\pi(s, s')$, is defined as the expected discounted number of visits to one particular state $s'$ if an agent starts in $s$ and follows $\pi$. Therefore, it is also possible to interpret SCs as a generalization of the SR. We formalize this notion in the following observation:

**Observation 1.** *Let $M$ be an MDP with a discrete state space $\mathcal{S}$. For a given policy $\pi$, let the Successor Representation be $\psi^\pi(s, s') = \mathbb{E}_\pi\left[\sum_{k=0}^\infty \gamma^k \mathbb{1}_{(S_{t+k+1}=s')} \mid S_t = s\right]$. By slight abuse of notation, the SC for a cluster $C_i$ is $\psi^\pi(s, C_i) = \mathbb{E}_\pi\left[\sum_{k=0}^\infty \gamma^k \mathbb{1}_{(S_{t+k+1}\in C_i)} \mid S_t = s\right]$. Therefore, we have that $\psi^\pi(s, C_i) = \sum_{s'\in C_i} \psi^\pi(s, s')$. It follows that as the number of clusters increases ($d \to |\mathcal{S}|$), the SCs and the SR of a policy become equivalent, i.e., as $|C_i| \to 1$ and $C_i \to \{s_i\}$, $\psi^\pi(s, C_i) \to \psi^\pi(s, s_i)$.*

Observation 1 implies that we can interpret SCs as a form of *spatial generalization* of the SR, modulated by the number of clusters, $d$. This result also extends to continuous states by replacing the summation with integrals, e.g., $\psi^\pi(s, C_i) = \int_{s'\in C_i} \psi^\pi(s, s')ds'$.

Recall that our main goal is to perform zero-shot transfer. To accomplish this, SF-based methods learn a set of policies $\pi(a|s; \mathbf{w})$, for all $\mathbf{w}$, capable of maximizing any linear combination of the reward features (Section 2). Then, given any reward function $r$, the agent can follow, in a zero-shot manner, the policy $\pi(a|s; \mathbf{w}_r)$, where $r \approx \boldsymbol{\phi} \cdot \mathbf{w}_r$. When employing the SR instead of SFs, Alegre et al. (2022) showed that this process induces an optimal policy for any reward function. Since according to Observation 1 SCs are a generalization of the SR, they also benefit from similar properties. In Appendix A.2, we discuss this theoretical connection and show that SCs also produce an optimal set of policies as the number of clusters increases.

---

[4]This follows directly from the fact that with one-hot vector features, $\sum_{k=1}^d \phi_k(s) = 1$, and $\sum_{t=0}^\infty \gamma^t = \frac{1}{1-\gamma}$.

Another relevant property of the cluster-based reward features used by SCs is that they are orthonormal and, therefore, of full rank $d$. Touati et al. (2023) analysed the feature rank of different methods for learning reward features. They showed that methods that produce features with a rank lower than $d$ (which they refer to as feature collapse) have lower zero-shot performance. We note that independent and orthogonal features such as ours are likely easier to approximate and predict by function approximators, especially in the context of SFs.

## 4 A Practical Successor Clusters Algorithm for Zero-Shot Transfer

In this section, we introduce a practical algorithm, based on the definition of SCs and their theoretical motivations presented in the previous section, for unsupervised zero-shot transfer in RL. In particular, we introduce an offline deep RL algorithm that learns SCs in an unsupervised manner (i.e., without information on the distribution of testing tasks) using only a dataset of offline experiences $\mathcal{D} = \{(s_i, a_i, s'_i)\}_{i=1}^n$. In summary, given a dataset of experiences $\mathcal{D} = \{(s_i, a_i, s'_i)\}_{i=1}^n$, our method learns *(i)* a USFA modeling the Successor Clusters, $\boldsymbol{\psi}(s, a; \mathbf{w})$, and a corresponding policy network $\pi(s; \mathbf{w})$ (Section 4.1); and *(ii)* a latent space projection function, $\varphi(s)$, in conjunction with a goal-conditioned action-value function, $Q(s, a; g)$, with the latter guiding the construction of the former (Section 4.2). Reward features $\boldsymbol{\phi}(s) = [\mathbb{1}_{C_1}(\varphi(s)), \cdots, \mathbb{1}_{C_d}(\varphi(s))]$ are constructed using online K-Means (Section 4.2). All above-mentioned components are learned as the agent optimizes reward functions of the form $r_\mathbf{w} = \boldsymbol{\phi} \cdot \mathbf{w}$, with weights $\mathbf{w}$ sampled (without loss of generality) from $\mathcal{W} = \{\mathbf{w} \in \mathbb{R}^d \mid \|\mathbf{w}\|_2 = \sqrt{d}\}$ (Section 4.3). The algorithm's pseudocode is presented in Algorithm 1 and is discussed in detail in the following sections.

---

**Algorithm 1:** Offline Successor Clusters

**Input:** Dataset of offline transitions, $\mathcal{D} = \{(s_i, a_i, s'_i)\}_{i=1}^n$

**1** Initialize the following models: Successor Clusters $\boldsymbol{\psi}(s, a; \mathbf{w})$ (Section 4.1); policy $\pi(s; \mathbf{w})$; reward features $\boldsymbol{\phi}(s)$; projection function $\varphi(s)$ onto the latent space $\bar{\mathcal{S}}$ (Section 4.2); and goal-conditioned action-value function $Q(s, a, g)$ (Section 4.2).

**2** **while** *Training budget lasts* **do**

**3**     Sample minibatch of transitions $\{(s_i, a_i, s'_i)\}_{i=1}^b \sim \mathcal{D}$ and weight vectors $\{\mathbf{w}_i\}_{i=1}^b \sim \mathcal{W}$ (Section 4.3)

        ▷ Update goal-conditioned $Q$ and latent features $\varphi$. Recompute clusters $\phi$

**4**     **Update** goal-conditioned action-value function $Q(s, a; g)$ via a goal-conditioned RL method

**5**     **Update** latent space projection function $\varphi$ with $L_\varphi$ (Equation (8))

**6**     **Update** cluster features $\boldsymbol{\phi}$ using datapoints $\varphi(s)$ via online K-Means

        ▷ Update SCs $\psi$ and policy $\pi$

**7**     **Update** $\boldsymbol{\psi}$ (Equation (7)) and $\pi$ (Section 4.1)

---

### 4.1 Training the USFA

Since we would like to learn SCs that generalize over a range of different reward functions (each encoded by a different weight vector $\mathbf{w}$), we employ a USFA $\boldsymbol{\psi}(s, a; \mathbf{w}) \approx \boldsymbol{\psi}^{\pi_\mathbf{w}}(s, a)$, approximated via a neural network. Recall that **Successor Clusters $\boldsymbol{\psi}(s, a; \mathbf{w})$ represent the expected discounted time spent in each cluster when solving a given task $\mathbf{w}$** by following policy $\pi(s; \mathbf{w})$. In discrete-action domains, the policy $\pi(s; \mathbf{w})$ can be directly inferred from the SCs: $\pi(s; \mathbf{w}) \in \arg\max_{a \in \mathcal{A}} \boldsymbol{\psi}(s, a; \mathbf{w}) \cdot \mathbf{w}$. A USFA can be trained with DQN-like updates:

$$\mathcal{L}_{\boldsymbol{\psi}} = \mathbb{E}_{\substack{s,a,s' \sim \mathcal{D} \\ \mathbf{w} \sim \mathcal{W} \subset \mathbb{R}^d}} \left[ \|\boldsymbol{\phi}(s) + \gamma \boldsymbol{\psi}(s', a^\star; \mathbf{w}) - \boldsymbol{\psi}(s, a; \mathbf{w})\|_2^2 \right], \quad \text{where } a^\star = \pi(s'; \mathbf{w}) \in \arg\max_{a' \in \mathcal{A}} \boldsymbol{\psi}(s', a'; \mathbf{w}) \cdot \mathbf{w}.$$

$$(7)$$

In continuous-action domains, the actor policy $\pi(s; \mathbf{w}) \in \arg\max_{a \in \mathcal{A}} \boldsymbol{\psi}(s, a; \mathbf{w}) \cdot \mathbf{w}$ can be optimized via deterministic policy gradients, as in the TD3 algorithm (Fujimoto et al., 2018). The proposed approach for sampling tasks $\mathbf{w}$ in $\mathcal{W} \subset \mathbb{R}^d$ will be detailed in Section 4.3.

Note that the agent can be in only one cluster at a time: $\phi$ is a one-hot vector, so $\sum_{k=1}^{d} \phi_k(s) = 1$. This means that for a fixed discount $\gamma$, the total expected discounted "visit time" given by the Successor Cluster representation is constant: for any $s, a$ and $\mathbf{w}$, $\sum_{k=1}^{d} \psi_k(s, a; \mathbf{w}) = \mathbb{E}_{\pi_{\mathbf{w}}}[\sum_{t=0}^{\infty} \gamma^t] = 1/(1-\gamma)$. We can exploit this property to constrain the USFA neural network to only produce outputs that satisfy this property. For instance, an agent should not be able to predict it will spend 150% of its time in cluster $C_1$, or $-20\%$ in cluster $C_2$; outputs should always be positive and sum up to 100%. This can be achieved by adding a rescaled softmax layer after the final layer of $\psi$, constraining it to produce outputs whose sum is $1/(1-\gamma)$.

## 4.2 Learning the Minimal Path Distance Function

We now introduce a method for learning the state distance function, $d_\star$, introduced in Equation (5), to use when partitioning the state space. Note that this is a proper distance function and can be used for clustering the state space $\mathcal{S}$. However, it cannot be used with traditional clustering methods such as K-Means, as their convergence guarantees hold only for Euclidean spaces. To overcome this problem, we construct a latent space projection $\varphi : \mathcal{S} \to \bar{\mathcal{S}} \subset \mathbb{R}^n$ where, by construction, the latent Euclidean distance $\delta = \|\cdot\|_2$ between $\varphi(s_1)$ and $\varphi(s_2)$ matches with the average minimal path between $s_1$ and $s_2$ in the MDP, under the original distance function $d_\star$. Fouss et al. (2005) showed that, for undirected graphs, there exists a latent space where Euclidean distances in the latent space correspond to the square root of the average commute time in the graph. This allows us to define the following loss function to learn $\varphi$:

$$L_\varphi \triangleq \mathbb{E}_{s,g \sim \mathcal{D}}\left[(\delta(s,g) - d_\star(s,g))^2\right]. \tag{8}$$

Computing this loss only requires to sample pairs of states from the MDP. However, the crux of the effort is in learning $d_\star$ through goal-conditioned RL as per Equation (5). We rely on existing goal-oriented RL algorithms, especially the Hindsight Experience Replay (HER) (Andrychowicz et al., 2017) technique, to estimate $\max_a Q(s,a;g) \approx \max_\pi v_\pi(s;g)$. This leads to the final loss for $\varphi$,

$$L_\varphi = \mathbb{E}_{s,g \sim \mathcal{D}}\left[\left(\|\varphi(s) - \varphi(g)\|_2 - \frac{1}{2}\max_a Q(s,a;g) - \frac{1}{2}\max_a Q(g,a;s)\right)^2\right]. \tag{9}$$

Such goal-reaching agents can be trained entirely offline with architecture and algorithms virtually identical to a USFA, which limits hyperparameter search. We then compute a clustering $\{C_i\}_{i=1}^{d}$ of the states *in the latent space*, $\varphi(\mathcal{S})$, using K-Means. The resulting rewards features used in our method are then $\phi(s) = [\mathbb{1}_{C_1}(\varphi(s)), \cdots, \mathbb{1}_{C_d}(\varphi(s))]$, leading to the SCs method described in Section 3.4.

## 4.3 Reward Function Approximation and Task Vector Normalization

We want to compute the best linear approximation with weights $\mathbf{w}$ given a reward function $r$. Section 3.2 established that the optimal $\mathbf{w}$ for cluster occupancy features corresponds to average cluster rewards, $\mathbf{w} = [r(C_1), \cdots, r(C_d)]^T$, with each $r(C_i)$ defined as the average reward in the cluster $C_i$. More concretely, consider samples $\{s_j, r(s_j)\}_{j=1}^{n}$ from an arbitrary reward function $r$. Let $X_i$ be the set of states from the samples that belong to cluster $C_i$. Then, the weight vector $\mathbf{w}$ that best approximates this reward can be efficiently computed as the average reward per cluster:

$$w_i \approx \frac{1}{|X_i|} \sum_{s \in X_i} r(s), \ \forall i \in \{1, \cdots, d\}. \tag{10}$$

The fact that the task vector is expressed entirely in terms of the reward function is of particular relevance. Indeed, any reward function has two key properties that can be exploited: first, it is independent of scaling: the optimal policies do not change if the reward is scaled by a positive factor $\alpha$. Second, in a *continual environment* (with no terminal states), the reward function is independent of translation: the optimal policies do not change if the reward is translated by a constant $\beta$. We can use this to our advantage to prevent USFAs from learning "redundant" task vectors, i.e., vectors that share a same optimal policy. For

example, tasks $\mathbf{w} = [-1, 0, 1]$, $\mathbf{w} = [1, 2, 3]$ and $\mathbf{w} = [0, 0.33, 0.66]$ all share the same optimal policy. We opt to normalize all task vectors onto the sphere, $\sqrt{d}(\mathbf{w} - \bar{\mathbf{w}})/\|\mathbf{w}\|_2$, before passing it to the USFA network. That is, without loss of generality, we define $\mathcal{W} = \{\mathbf{w} \in \mathbb{R}^d \mid \|\mathbf{w}\|_2 = \sqrt{d}\}$. We also randomly sample task vectors from this sphere during training to simulate different reward functions.

### 4.4 Following a Provided Trajectory

The interpretable nature of SCs has one more key advantage: we can use it for imitation learning, i.e., to train an agent to imitate a given trajectory. In RL, reward design is a major problem since in many cases, designers can characterize the overall behavior the agent should produce, but creating a corresponding reward that induces this behavior is not trivial. The agent will often take shortcuts, find local optima, or find loopholes to ignore the desired objective entirely. With SCs, we can reason in terms of expected cluster visits or analyze a trajectory in terms of expected visits, which can, in turn, be mapped to a task vector $\mathbf{w}$. First, assume that a human expert wishes to generate a trajectory $\tau = s_0, s_1, \cdots s_T$. This can be mapped to total cluster visits occurring during the trajectory: $O(s_0) = \sum_{t=0}^{T} [\mathbb{1}_{C_1}(s_t), \cdots, \mathbb{1}_{C_d}(s_t)]$ (ignoring the discount $\gamma$ here for simplicity). We can alternatively assume the expert knows which clusters they would like the agent to visit, and that the associated visitation count is the diameter of the cluster. These desired visits can then be used to find the vector $\mathbf{w}$ with which the expected visits predicted by the SCs will be closest to the desired visits:

$$\mathbf{w} \triangleq \arg\min_{\mathbf{w}} \|\boldsymbol{\psi}(s_0, a_\star; \mathbf{w}) - O(s_0)\|^2, \tag{11}$$

where $a_\star = \arg\max_{a \in \mathcal{A}} \boldsymbol{\psi}(s, a; \mathbf{w}) \cdot \mathbf{w}$. This objective can be easily optimized with gradient descent. During this process, $\mathbf{w}$ can be constrained to be on the $\sqrt{d}$-radius sphere (see Section 4.3).

## 5 Experiments

In this section, our goal is to empirically evaluate the Successor Clusters method and to empirically validate its theoretical properties. In particular, we aim to investigate the following research questions: **Q1:** Do SCs result in better zero-shot performance when compared to other methods for learning reward features (Section 5.1)? **Q2:** Do SCs result in lower reward prediction error, as suggested by Theorem 1 (Section 5.2)? **Q3:** Can we generate interpretable visualizations of the different learned components of our method (Section 5.3)?

### 5.1 Evaluation of Zero-Shot Transfer Performance

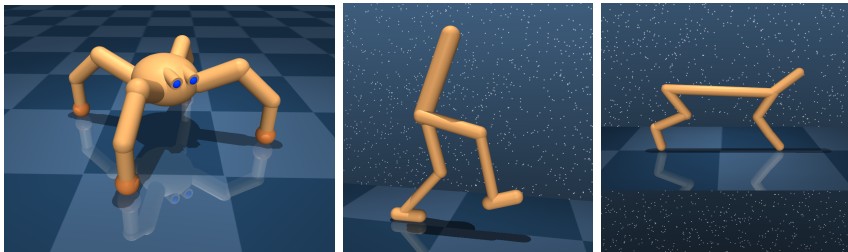

Figure 4: Domains used for evaluating zero-shot transfer performance: Quadruped, Walker, and Cheetah robots from URLB (Laskin et al., 2021). Transfer tasks involve performing movements such as walking, running, jumping, and flipping.

We follow the experimental setting employed by Touati et al. (2023), which comprises robotic tasks from the Unsupervised RL Benchmark (URLB) (Laskin et al., 2021). In this fully offline setting, agents have access to an offline replay buffer containing $5 \times 10^6$ experiences collected using an exploration policy trained with Random Distillation Network (Burda et al., 2019) as provided by the ExORL repository (Yarats et al., 2022). Agents are then tasked with learning reward features and SFs in a purely unsupervised manner with a budget of $10^6$ gradient updates. The method's hyperparameters and other experimental details are specified in Appendix C. We evaluate the zero-shot performance of each method in the Quadruped, Walker, and

Cheetah simulated robots from URLB, depicted in Figure 4. The performance is evaluated w.r.t. different reward functions that the agent approximates using their learned features (Section 4.3), e.g., rewards for walking, standing, and jumping. All reward functions are described in Appendix C.2.

We compare Successor Clusters with the Laplacian representation (Wu et al., 2019), since this is the approach that generated the best overall zero-shot performance when used with SFs in the extensive evaluations conducted by Touati et al. (2023). Laplacian eigenfunctions provide a Fourier-like basis that span value functions of an MDP, as detailed in Section 3.1. We also compare employing SCs using Laplacian features as the latent space $\varphi$ used to define the *distance function* used for clustering instead of the average minimal path (Minpath for short) introduced in Section 3.3, as well as Minpath latent space features for completeness. We used $d = 50$ as the dimension of the reward features $\phi(s) \in \mathbb{R}^d$, similarly to what is done in Touati et al. (2023). Recall that for SCs, $d$ corresponds to the number of clusters. In which follows, we normalize returns and show the percentage of the performance w.r.t. the best-encountered return over all agents on each task. We depict the mean and 95% stratified bootstrap confidence intervals over ten runs with different random seeds (Agarwal et al., 2021).

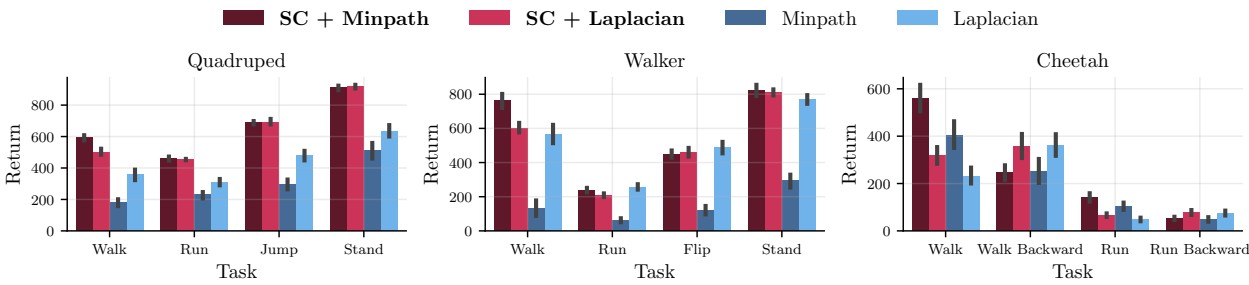

Figure 5: Mean returns obtained in the Quadruped, Walker, and Cheetah robotic tasks of the URLB when using Successor Clusters (SC); or method, depicted in shades of red, or Successor Features (SF); depicted in shades of blue, with Minpath or Laplacian reward features.

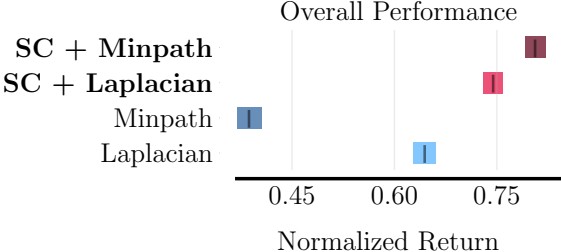

Figure 6: Overall performance (mean normalized returns with 95% stratified bootstrap confidence intervals) in the Quadruped, Walker, and Cheetah robotic tasks depicted in Figure 5.

Figure 7: Ablation of the dimension $n \in \{25, 50, 100, 200\}$ of the Minpath latent space $\varphi(s) \in \mathbb{R}^n$ used by SCs in the four tasks of the Walker domain.

In Figure 5, we show the mean return of each agent obtained in each task and robot. We note that agents using SCs (red bars) very often outperformed their "raw" features counterparts (blue bars). Agents using solely Minpath reward features, in general, did not perform well in this benchmark. This is to be expected since they were not specifically designed to linearly approximate reward functions. Additionally, we show, in Figure 6, the overall performance of each method averaged over all tasks. Notice that our method, SC + Minpath, not only obtained the best overall performance, but that combining SCs with other types of reward features (e.g., SC + Laplacian) results in improvement when compared to using the original reward features directly (Laplacian). These results imply a positive answer to research question **Q1** and show that SC's theoretical properties (as motivated by Theorem 1) ensure that agents perform better when tackling tasks with previously unseen rewards in a zero-shot manner.

In order to investigate the impact of the dimension $n$ of latent vectors $\varphi(s) \in \mathbb{R}^n$, we conduct an ablation experiment in the Walker task whose results are depicted in Figure 7. Notice that using too low values of $n$ (e.g., 25) hurts performance, possibly due to the dimensionality not being sufficient to encode distances in the high-dimensional state-space of the Walker domain. Similarly, too high values (e.g., 200) led to decreased performance. The best performance was found for intermediate values of $n$, between 50 and 100.

Additionally, in Appendix C.3, we include a comparison with the Forward-Backward (FB) representation method (Touati et al., 2023), a state-of-the-art approach for unsupervised zero-shot transfer in RL. While our primary objective is to improve upon Successor Feature-based methods by designing more effective reward representations, our approach also performs competitively when compared to FB. Importantly, please note that FB adopts a fundamentally different strategy for zero-shot learning and does not belong, strictly speaking, to the class of techniques we aim to improve upon. Specifically, unlike our method, *(i)* it does not leverage a linear decomposition of reward functions; and *(ii)* it focuses on learning features that support the reconstruction of optimal $q$-functions, rather than features that enable accurate linear approximations of a broad range of reward functions.

## 5.2  Evaluation of Reward Approximation Quality

To investigate **Q2**, and **Q3**, we perform qualitative and quantitative experiments on the Maze environment, as it allows us to more easily visualize the agent's behaviors. We did not include this domain in the validation of **Q1** since all algorithms had an identical (near-optimal) performance on the goal-reaching tasks defined in this benchmark. The Maze state space consists of the location of the agent in a 2D square space $(x, y)$ as well as its velocity along each dimension $(\dot{x}, \dot{y})$, for a total of 4 inputs. There is a cross-shaped wall in the middle of the environment (shown in black on all figures), leading to non-trivial geodesic distances. Actions are 2D vectors defining the acceleration of the agent along the $(x, y)$-plane. We follow the assumption from Touati et al. (2023) that possible reward functions depend on the agent's position and are invariant with respect to its speed, that is, the family of possible rewards is $\Gamma = \{r \mid r(x, y, \dot{x}, \dot{y}) = r(x, y)\}$. Clusters are defined over the 2D position plane. The environment has a long horizon: it takes around 400 steps for an agent to move from one corner of the square to the other.

We start by comparing the reward prediction error of SCs and Laplacian w.r.t. the reward coverage maximization objective (Equation (3)) considering different families of reward functions. We also study the impact of the hyperparameter $d$, the number of reward features, on each of the methods. We evaluate the agent using four different families of reward functions, each of which represents relevant tasks that an agent could be faced with during transfer. The reward families are *(i)* goal-reaching rewards, each a single highly-truncated Gaussian placed at the 20 goal locations from the benchmark; *(ii)* trajectory rewards, which follow a path along the MDP with increasing rewards leading up to some goal state; *(iii)* Gaussians, a sum of 2 to 20 Gaussians with random means, standard deviations, and weights and *(iv)* smooth functions as combinations of linear, trigonometric and exponential functions. We show a visualization of all these reward functions in Appendix C.5. For all of these reward families, we compute the best linear approximation $\mathbf{w}$ as $\min_{\mathbf{w}} \sum_{\mathcal{S}} (r(s) - \phi(s) \cdot \mathbf{w})^2$ and show the resulting mean squared error (MSE) in Figure 8.

In Figure 8, notice that SCs lead to a significantly lower prediction error when compared to Laplacian features when evaluated in all families of reward functions. Although the goal-reaching reward functions are not Lipschitz, their discontinuities are local, so the performance of SCs is not significantly impacted. This is an important observation: although Theorem 1 holds for Lipschitz rewards, the piecewise nature of the SCs reward approximation makes it a strong option for sparse rewards. In comparison, Laplacian features consist only of smooth functions, so the corresponding approximation struggles with such sparse rewards. We further discuss the Lipschitz assumption and the consequences of violating it in Section 7.1. As discussed in Section 3.2, our cluster-based reward features are guaranteed to approximate reward functions increasingly better as the number of features, $d$, increases. In this Figure 8, we empirically demonstrate this property, and observe that it does not hold for the Laplacian-based features. This may be due to the fact that latter eigenfunctions correspond to "high-frequency" functions.

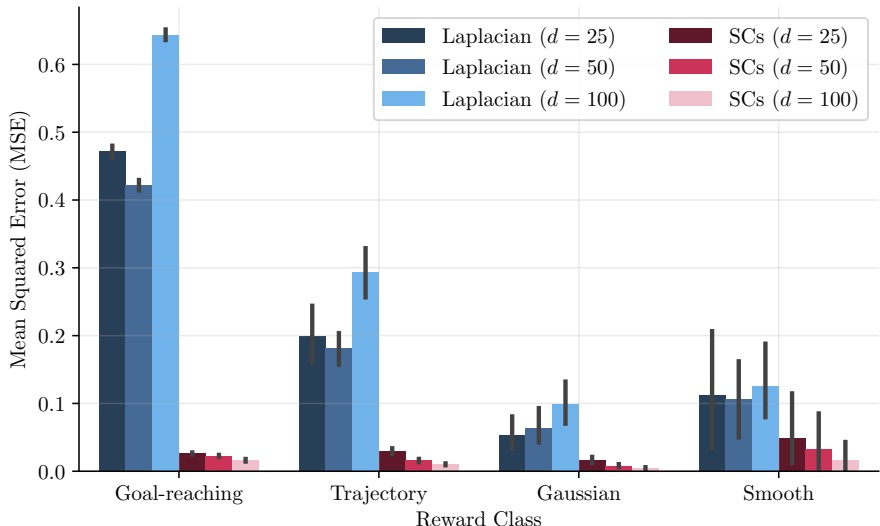

Figure 8: Mean squared error (MSE) of the reward approximation of SCs and Laplacian for various families of reward functions and a varying number of reward features, $d$.

### 5.3 Evaluation of Successor Clusters as an Interpretable Representation of Behaviors

Finally, we investigate **Q3** based on the interpretability associated with our method by providing visualizations of the learned latent space $\varphi$, learned clusters $\{C_i\}_{i=1}^d$, and the SCs predictions given by the USFA network $\boldsymbol{\psi}(s, a; \mathbf{w})$ in the Maze domain.

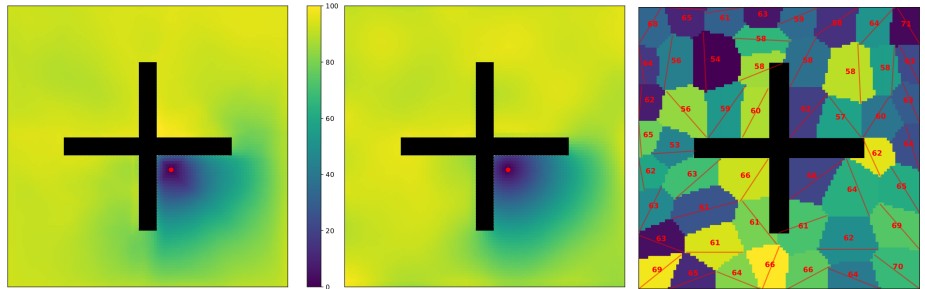

Figure 9: Left: Learned goal-reaching $Q$-values, $Q(\cdot; g)$, for a given goal state $g$ (red circle). Middle: Associated latent distance, $\|\varphi(s) - \varphi(g)\|_2$. Right: Partitioning of the space into $d = 50$ clusters with associated diameters depicted in red.

**Minimal Path Latent space.** As discussed in Section 4.2, our method (Algorithm 1) trains a goal-reacher policy and use its $Q$-value to construct a latent space representation where the distance between any two states matches the minimal number of time steps necessary to transition from one to the other (Minpath distance). In Figure 9, we visualize the $Q$-values, $Q(\cdot; g)$, associated with reaching a goal state $g$ and the corresponding distance in the latent space, $\|\varphi(s) - \varphi(g)\|_2$. Notice that the distance function is learned effectively since it correctly accounts for topological properties of the environment, such as the geodesic distances induced by the presence of obstacles. Figure 10 shows t-SNE visualizations of the latent space, similarly as done by Touati et al. (2023). We can observe that the latent space learned by Minpath is consistent with the structure of the environment's state space, with states separated by obstacles being projected onto corresponding distance embeddings. In comparison, the learned Laplacian representation does not result in an easily interpretable visualization of the MDP dynamics.

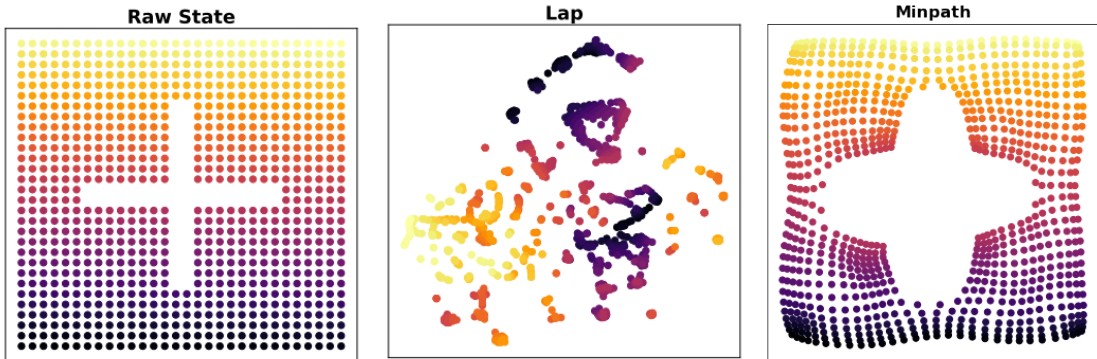

Figure 10: t-SNE visualization of the learned latent space (Raw state, Laplacian, Minpath).

**Clustering.** In Figure 9 (rightmost panel), we show a visualization of the learned clustering over the Maze state space. Although the state distribution in the offline replay buffer is strongly biased towards the upper-right region of the maze (where initial states are located), the partitioning of the state space is not significantly biased, and the various regions of states are fairly evenly covered. We also show the radii of each cluster—most clusters have a diameter of around 70 time steps, showing that Maze is quite a long-horizon environment. The topology of the environment is also captured by the structure of clusters, as the clusters are located around obstacles and do not overlap with them.

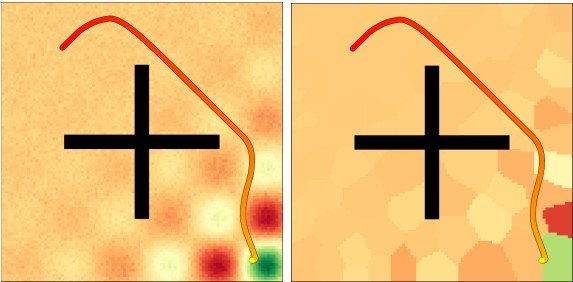

(a) Left: Reward function being approximated (ground truth). Right: Associated reward approximation. The trajectory executed by following the agent's policy is shown in red-yellow.

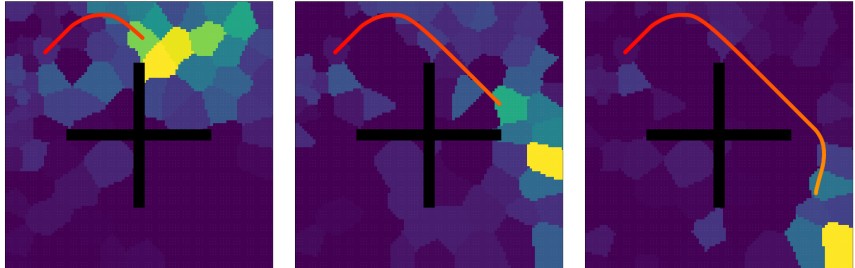

(b) Predicted cluster occupancy induced by the reward approximation after three steps along the trajectory following the agent's policy (red-yellow curve).

Figure 11: Visualization of the reward function approximation and the cluster occupancy predicted by SCs.

**Successor Cluster prediction.** We now show a visualization of one of the central elements of this paper: the cluster visitations predicted by the SCs. In Figure 11, we visualize the complete zero-shot transfer process implemented by our method. First, in Figure 11a, an arbitrary reward function $r$ is discretized into clusters by identifying the weight vector $\mathbf{w}$ that best approximates it, i.e., $r \approx \boldsymbol{\phi} \cdot \mathbf{w}$, following the

method introduced in Section 4.3. Next, the SCs USFA predicts the future cluster occupancy of the optimal policy w.r.t. this reward, and follows the corresponding trajectory (Figure 11b). Since Maze has a long horizon, the agent only predicts the next few clusters it will visit due to small values of $\gamma^t$ as $t$ increases. We provide an animated version of this prediction, as the agent advances through the maze, at https://sites.google.com/view/successor-cluster-vis/accueil. This prediction can be used to enable or improve several human-AI feedback loops. First, from a research and development perspective, users can have a better understanding of the network's error and more easily detect implementation mistakes. Second, for safe RL applications, we can study the agent's expected visit of undesirable clusters and adapt the reward function and training accordingly. Third, this also allows the users to better understand the intent of the agent and the impact of different tasks $\mathbf{w}$ on the generated trajectory, without needing to deploy the policy. Finally, the intuitive nature of cluster predictions makes it a better interface for users than commonly used black-box features.

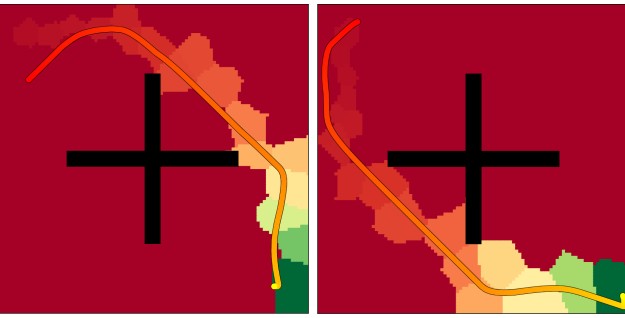

Figure 12: Controlling a SCs agent by changing cluster reward weights, $\mathbf{w}$, to produce a trajectory along the top or bottom regions of the maze (left and right panels, respectively) and that reaches a target cluster in the bottom-right region of the environment.

**Controlling the agent.** Our method's USFA is trained to *navigate* to various clusters, that is, the agent is able to traverse or avoid clusters $C_i$ based on their corresponding individual values, $w_i$. By adjusting $\mathbf{w}$, we can thus control the expected trajectory executed by the agent over clusters of states. We visualize this capability in Figure 12. Here, different choices of $\mathbf{w}$ control whether the agent should navigate along the top or bottom of the maze when reaching a target cluster in the bottom-right region of the environment.

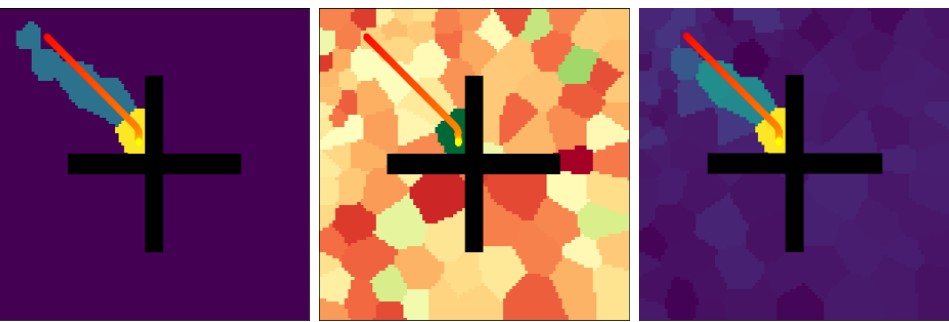

Figure 13: Using SCs to induce agents to follow a desired trajectory. Left: Specification of the time a designer would like the agent to spend in each cluster. Middle: The $\mathbf{w}$ vector optimized to generate these visits from the USFA. Right: The effective USFA predictions along with the agent's generated trajectory.

**Fitting a desired trajectory.** Finally, as discussed in Section 4.4, our method allows for a given a trajectory of interest to be expressed in terms of desired cluster visits: $O(s_0) = \sum_{t=0}^{T} [\mathbb{1}_{C_1}(s_t), \cdots, \mathbb{1}_{C_d}(s_t)]$. Then, it is able to identify a task vector $\mathbf{w}$ that expresses a reward function that induces the desired trajectory. While this enables agents to perform behavior cloning, one can also provide a trajectory that the designer does not know how to achieve. In Figure 13, we employ this idea to induce an agent to reach the

closest corner of an obstacle depicted in yellow (leftmost panel). This results in a trajectory where the agent spends a few steps in each of the clusters along the way towards the obstacle, and the remaining steps of the trajectory in the corner cluster. This gives us the desired visits from the starting state, $O(s_0)$. Next, our method computes $\mathbf{w}$ through Equation (11) via gradient descent. This process identifies the vector $\mathbf{w}$ that is locally most likely to result in the desired cluster visits, $\boldsymbol{\psi}(s_0, a_\star; \mathbf{w}) \approx O(s_0)$. Finally, we can then execute the zero-shot policy for $\mathbf{w}$ to obtain the desired trajectory. Note again that this procedure is qualitatively different from behavior cloning. In particular, we only need to provide approximate cluster occupancies, but we do not need to inform the agent about what actions it should take.

## 6 Related Work

In this section, we discuss existing relevant approaches for learning reward features, as well as other related works in the literature on successor features, discrete representations, and unsupervised skill discovery.

**SFs and reward feature learning.** Previous works have tackled the problem of learning a set of reward features to be used within the SFs framework. Given a set of training tasks, Barreto et al. (2017a; 2018); Carvalho et al. (2023b;a) construct reward features via linear regression with the goal of linearly approximating the reward functions in this training set. This type of approach works well when a set of representative training tasks is available for training, with the assumption that test tasks share structural similarities with training tasks. We, by contrast, tackle the *unsupervised setting* in which an agent has to learn reward features without access to a predetermined distribution of tasks. Hansen et al. (2020) proposed to learn features via a behavior mutual-information objective (Eysenbach et al., 2019b) in which the weight vector plays both the role of the "skill" vector (typically employed in mutual information RL algorithms) and the role of the linear reward weights. Touati & Ollivier (2021); Touati et al. (2023) introduced the forward-backward representation, which models a low-rank approximation of the successor measure of a policy. Although these approaches work well in practice, they do not produce representations that are naturally interpretable, such as the ones that underlie our method. Reinke & Alameda-Pineda (2023) studied the problem of extending SF-based methods to solve tasks defined as non-linear functions of reward features. However, they assumed that the reward features were provided *a priori* and did not tackle the problem of learning expressive reward features. Another approach designed to optimize reward coverage maximization, although not under the SFs framework, was introduced by Frans et al. (2024) and consists in approximating randomly sampled reward functions. However, this approach is very computationally intensive. Moreover, it requires a manually-designed specification of the distribution of training reward functions, for which no theoretical guarantees are given with respect to their ability to generalize to arbitrary test reward distributions. We, by contrast, characterize and construct a particular choice of $\phi$ and $\mathbf{w}$ with strong relevant theoretical properties. Other works have explored the problem of learning a set of optimal policies given predefined reward features (Zahavy et al., 2021; Alver & Precup, 2022a; Alegre et al., 2022)—an important problem orthogonal to the one investigated in this paper.

**Unsupervised skill discovery (USD).** USD methods aim to learn a set of policies that can be used to solve unseen downstream tasks. USD methods generally try to create policies that reach key states (Machado et al., 2018; Jinnai et al., 2019) or that maximize skill diversity through a mutual information objective (Gregor et al., 2017; Eysenbach et al., 2019b). In the context of Successor Features, Hansen et al. (2020) introduced state features associated with the probability of each skill visiting a given state. Eysenbach et al. (2022) have shown, in a geometric study of USD, that diversity-based methods ignore regions and associated skills that are too close to the sampling distribution, since these may be difficult to differentiate. In comparison, SCs allows agents to evenly reach the entire state space being approximated by state partitioning.

**Discrete representations.** Grimm et al. (2019) proposed using a set of *disentangled* features that can be controlled independently by discretizing domain features into a set of discrete bins. However, these can only be employed in domains in which it is possible to specify tasks in terms of disentangled transformations. Moreover, they assume that the order of actions executed by the agent does not matter, which is typically not true in RL. Ramesh et al. (2019) also employ a clustering approach in the context of the successor representation framework. However, their approach is significantly different than ours. They, for instance,

cluster the state space using the SR as a distance metric and use the clusters as sub-goals to define options. We, by contrast, cluster the state space w.r.t. the minimal path between states and use clusters to define reward features used to construct SFs. Our work also shares interesting connections with reward-predictive representations (Lehnert et al., 2020; 2022). In particular, Lehnert et al. (2022) defined reward-predictive clustering as a state abstraction that groups states predicted to produce the same future reward sequences. Although they show that their learned representation (given a training task) can be used as an initial solution to accelerate the agent's adaptation to novel reward functions, the proposed framework is not capable of performing zero-shot transfer. We, by contrast, construct a reward clustering representation tailored for zero-shot adaptation. Farebrother et al. (2023) define successor measures with indicator functions in a way that resembles SCs (Equation (6)). However, they use random indicator functions (or random clusters) with the goal of deriving auxiliary tasks in a single-task setting. We, instead, learn clusters with properties relevant to the goal of approximating reward functions in the multi-task zero-shot setting. Similarly to our reasoning, they argue that binary features can be beneficial since they are easier to adjust compared to real-valued functions. Another form of discretization used in RL with function approximation is tile coding (Sherstov & Stone, 2005; Sutton & Barto, 2018). In tile coding, the state space is split into regions visited under an indicator function. Our method has key differences when compared with tile coding: *(i)* the amount of features in $\phi$ is strongly limited, making it difficult to layer tile maps; *(ii)* the bounds of the observation space are generally not known in advance (or are very large), making it difficult to define tiles; *(iii)* with limited features, off-the-shelf tile coding may tile regions of $\mathbb{R}^d$ that are not part of the state space $\mathcal{S}$. This is a limitation avoided by directly clustering states in $\mathcal{S}$, as in our work.

# 7 Conclusion

We now discuss a few limitations of our approach, and then summarize our theoretical contributions, empirical findings and comment on relevant future research directions.

## 7.1 Limitations

**Curse of dimensionality.** In high-dimensional spaces, the required number of clusters, $d$, can be high due to the curse of dimensionality. From a practical standpoint, it is important for all prior knowledge about the reward space of interest, $\Gamma$, to be leveraged to reduce the space being partitioned. From a theoretical perspective, we expect it should be possible to borrow ideas from the tile coding literature to layer *cluster maps*. In particular, with $d$ binary feature maps, one should be able to define up to $2^d$ regions of the state space. Practical algorithms for achieving this (and learning the associated USFAs) could substantially improve the scalability of Successor Clusters.

**Lipschitz reward functions.** A main limitation of our method is generalizing over non-smooth reward functions. We restrict our theoretical analysis (Theorem 1) to the space of Lipschitz reward functions. We now briefly discuss how to relax this constraint to *local Lipschitz functions*. Let $r$ be locally Lipschitz on $\mathcal{S} \setminus \check{\mathcal{S}}$ where $\check{\mathcal{S}}$ is the set of states in the region where the Lipschitz property is not satisfied. For clusters $C_{\mathcal{S}}$ that have no state in $\check{\mathcal{S}}$, the guarantee of Theorem 1 remains. For clusters $C_{\check{\mathcal{S}}}$ that contain states of $\check{\mathcal{S}}$, the guarantee is lost. In sparse reward environments, $\check{\mathcal{S}}$ is quite small, which is why the method still performed well on the goal-reaching tasks (Figures 8 and discussion in Section 5.3). With Successor Clusters, this loss of guarantee on $\check{\mathcal{S}}$ is interpretable under the observation that the policy cannot fine-tune within clusters. It might then either underestimate a cluster, avoiding a high-value part of the cluster because of the low-valued rest, or overestimate it, crossing a low-valued part of the cluster because of its higher average value. To the best of our understanding, non-Lipschitz reward functions that are not sparse are quite rare, so SCs should generally be able to produce a strong zero-shot policy.

**Exploration and sampling state distribution.** All the experiments in this paper were performed in an offline RL setting using the RND exploration policy dataset provided by URLB (Laskin et al., 2021). Offline RL is fundamentally limited by the quality of the dataset: not much can be learned about regions of the state space that are not covered in the dataset. For example, in the Maze domain, some exploration policies never make it past the first room. Hence, we should not expect zero-shot methods to be able to

reach there. However, Successor Clusters are dependent on the sampling distribution in clearer ways. In Theorem 1, our sum is defined over the state space, but in practice, we only have access to an expectation over the (unknown) sampling distribution. Concretely, this means that parts of the state space that have a higher density than others will produce more clusters, leading to imbalances in the state space partitioning and zero-shot policy performance across the state space.

## 7.2 Summary and Future Work

In this work, we introduced Successor Clusters (SCs), a novel method to tackle unsupervised zero-shot RL problems. Our approach leverages a novel mathematical formulation to the problem of learning reward features capable of spanning *families* of reward functions, and that are (by design) tailored for use within the SFs framework. We formally characterized how reward features can be constructed by partitioning the state space and mathematically quantified upper bounds on their capability to approximate any Lipschitz reward function. To construct partitions of clusters, we introduced a novel notion of distances in an MDP based on the minimal number of time steps to transition between pairs of states. We showed that such distance metrics can be learned via goal-conditioned reinforcement learning methods. By employing *cluster occupancy features*, SCs are capable of predicting the expected discounted time that a policy spends in each state cluster. After a pre-training phase, our method is capable of approximating and maximizing any new reward function in a zero-shot manner, i.e., without requiring any additional learning. Importantly, we formally showed that as the number and quality of clusters improve, the set of policies that can be induced by SCs converges to a set containing the optimal policy for *any* given task. We empirically demonstrated that our method outperforms a state-of-the-art competitor in challenging continuous control benchmarks, both in terms of zero-shot performance and reward approximation error. Additionally, we showed that our method naturally produces several *interpretable* components and can be used to visualize where an agent is likely to visit while solving a task. We believe many promising future research directions could be investigated. For instance, SCs could be extended to the hierarchical setting (Machado et al., 2023) by combining high-level SCs predictions and low-level goal-reaching policies. Another promising direction is combining SCs with model-based approaches (Bagot et al., 2023; Alegre et al., 2023). Since model-based techniques excel at short-term planning, they can potentially be used to fine-tune behaviors of SCs agents inside state clusters. We discuss these and other relevant future research ideas in more detail in Appendix B.

### Acknowledgments

This research received funding from the Flemish Government under the "Onderzoeksprogramma Artifcile Intelligentie (AI) Vlaanderen" programme. It was also supported by the euROBIN (European ROBotics and AI Network) [No. 101070596]. This study was financed in part by the Coordenação de Aperfeiçoamento de Pessoal de Nível Superior – Brasil (CAPES) – Finance Code 001.

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

## A    Theoretical Results

In this appendix, we show the proofs of the theorems introduced in this paper, as well as some additional theoretical results.

### A.1    Partitioning for Reward Coverage - Proof of Theorem 1

**Theorem 1.** *Let $\mathcal{C} = \{C_i\}_{i=1}^d$ be a partitioning of the state space $\mathcal{S}$ and let $\Gamma_{K\text{-}Lip}$ be the space of $K$-Lipschitz reward functions for some distance metric $\delta$ over $\mathcal{S}$. Let $\phi(s) = [\mathbb{1}_{C_1}(s), \cdots, \mathbb{1}_{C_d}(s)]^\top$ be the feature vector indicating subset occupancy: $\mathbb{1}_{C_i}(s) = 1$ if $s \in C_i$ and $0$ otherwise. Then, reward coverage maximization is bounded as follows:*

$$\mathbb{E}_{r \sim \mathcal{P}(\Gamma_{K\text{-}Lip})}\left[\min_{\mathbf{w} \in \mathbb{R}^d} \sum_{s \in \mathcal{S}} (r(s) - \phi(s) \cdot \mathbf{w})^2\right] \leq K \sum_{i=1}^d \frac{1}{|C_i|} \sum_{x,y \in C_i} \delta(x,y). \tag{12}$$

*Proof.* We are looking for $\phi : \mathcal{S} \mapsto \mathbb{R}^d$ such that $\mathbb{E}_{r \sim \mathcal{P}(\Gamma_{K\text{-Lip}})}\left[\min_{\mathbf{w}} \sum_{s \in \mathcal{S}} (r(s) - \phi(s) \cdot \mathbf{w})^2\right]$ is minimal, with $r : \mathcal{S} \mapsto \mathbb{R}$ some state-conditioned reward function and $\mathbf{w} \in \mathbb{R}^d$ some weight vector solving the linear least-squares problem for a given $r$ and $\phi$. Let $\phi(s) = [\mathbb{1}_{s \in C_1}, \cdots, \mathbb{1}_{s \in C_d}]^\top$ be the (one-hot) indicator vector encoding the location of a state within a partition $\{C_i\}$ of $\mathcal{S}$. Note that $\cup_{i=1}^d C_i = \mathcal{S}$ and $C_i \cap C_j = \emptyset \ \forall i, j$, by definition of a partition. Under this choice of $\phi$, we can solve the linear problem in $\mathbf{w}$ analytically:

$$\sum_{\mathcal{S}} (r(s) - \phi(s) \cdot \mathbf{w})^2 = \sum_{i=1}^d \sum_{x \in C_i} (r(x) - w_i)^2$$

To minimize over $\mathbf{w}$, we can cancel the derivative w.r.t $w_i$ and solving for $w_i$:

$$\frac{\partial}{\partial w_i} \sum_{i=1}^d \sum_{x \in C_i} (r(x) - w_i)^2 = \sum_{x \in C_i} \frac{\partial}{\partial w_i} (r(x) - w_i)^2 = 0$$

$$\sum_{x \in C_i} (r(x) - w_i) = 0$$

$$\implies w_i = \frac{1}{|C_i|} \sum_{x \in C_i} r(x)$$

So the optimal $w_i$ corresponds to the average reward on cluster $C_i$, and the whole $\mathbf{w}$ discretizes the reward function on the state space $\mathcal{S}$. Let $r$ be $K$-Lipschitz with respect to $\delta$ in the state space and the squared distance in the output space, i.e. $\forall s_1, s_2 \in \mathcal{S}$,,

$$\left(r\left(s_1\right) - r\left(s_2\right)\right)^2 \le K\delta\left(s_1, s_2\right).$$

We know from the prior computation of the optimal $\mathbf{w}$ that for $x \in C_i$, $\boldsymbol{\phi}\left(x\right) \cdot \mathbf{w} = \frac{1}{|C_i|} \sum_{y \in C_i} r\left(y\right)$. Then,

$$\sum_{s \in \mathcal{S}} \left(r\left(s\right) - \boldsymbol{\phi}\left(s\right) \cdot \mathbf{w}\right)^2 = \sum_{i=1}^{d} \sum_{x \in C_i} \left(r\left(x\right) - \frac{1}{|C_i|} \sum_{y \in C_i} r\left(y\right)\right)^2$$

Noting that $\forall x$ and constant $c \ne 0, r(x) = cr(x)/c$, so $r\left(x\right) = \frac{1}{|C_i|} \sum_{y \in C_i} r\left(x\right)$,

$$= \sum_{i=1}^{d} \sum_{x \in C_i} \frac{1}{|C_i|^2} \left(\sum_{y \in C_i} \left(r\left(x\right) - r\left(y\right)\right)\right)^2$$

$$(\text{Jensen's ineq.}) \le \sum_{i=1}^{d} \frac{1}{|C_i|^2} \sum_{x \in C_i} |C_i| \sum_{y \in C_i} \left(r\left(x\right) - r\left(y\right)\right)^2$$

$$(\text{Lip. prop.}) \le K \sum_{i=1}^{d} \frac{1}{|C_i|} \sum_{x,y \in C_i} \delta\left(x, y\right).$$

$\square$

## A.2 Partitioning for Unsupervised Skill Discovery

### A.2.1 Problem Formulation

In the previous sections, we motivated the use of a clustering of the state space to tackle the problem of reward coverage maximization (Section 3.1). In this section, we further motivate this choice by showing that this clustering is a theoretically complete method for Unsupervised Skill Discovery (USD).[5]

Building a strong reward-agnostic skill set to use for down-stream tasks is the core challenge of USD. While the goal of USD is intuitive, its actual objective was only really formalized within a geometric study by Eysenbach et al. (2022) as the *vertex discovery problem*. By considering the discounted state occupancy distribution of any policy $\pi$, we can reason over the set of *achievable* occupancies. Eysenbach et al. (2022) shows that *(i)* all convex combinations of achievable occupancies are also achievable, therefore the achievable set is a *polytope* in the space of state marginal distributions, and *(ii)* for any reward function, there is a policy at the vertex of the polytope that maximizes this reward.

Therefore, it is enough to look for the vertices of the polytope; and the USD objective is framed as follows: "*given a Controlled Markov Process, find the smallest set of policies such that every vertex of the state marginal polytope contains at least one policy*".

Alegre et al. (2022) have shown that the vertex discovery problem is equivalent to finding the Convex Coverage Set (CCS) for state occupancy measures:

$$\text{CCS} = \left\{\rho^\pi \mid \exists \mathbf{w} \text{ s.t. } \forall \rho^{\pi'}, \rho^\pi \cdot \mathbf{w} \ge \rho^{\pi'} \cdot \mathbf{w}\right\}.$$

The CCS corresponds to the set of non-dominated policies over any given reward function $\mathbf{w} = \left[r_{\mathbf{w}}\left(s_1\right), \dots\right]^\top$; see the paper for further details. Critically, Alegre et al. (2022) further showed that the Successor Representation $\boldsymbol{\psi}^\pi$ corresponds to $\rho^\pi$, and provide an algorithm to learn the CCS with Successor Features, therefore

---

[5]The terms "skill" and "policy" are used interchangeably.

giving a solution to the vertex discovery problem. **If we can bind the state occupancy measure to its cluster approximation, then we can guarantee that the CCS we learn with Successor Clusters will get closer to optimal as the number of clusters grow.**

### A.2.2 Theorem and Proof

Let $\rho$ be a random state-marginal of the polytope, although the result we will show works on any partitioning over a distribution. Let $\varsigma = \{C_i\}$ be a set such that $\bigcup_i C_i = S$ and $\bigcap_i C_i = \emptyset$, i.e., $\varsigma$ is a partition of $S$ with clusters $C_i$. Define the cluster pseudo-marginal $\varrho(s \mid \varsigma) = \frac{1}{|C_i|} \sum_{s_i \in C_i} \rho(s_i)$, a discrete approximation of $\rho$ under clustering $\varsigma$. Note that since

$$\sum_s \varrho(s) = \sum_i \sum_{s \in C_i} \varrho_i = \sum_i \sum_{s \in C_i} \frac{1}{|C_i|} \sum_{s_i \in C_i} \rho(s_i)$$
$$= \sum_i \sum_{s_i \in C_i} \rho(s_i) = 1$$

and $\varrho \geq 0$, $\varrho$ can be seen as a distribution, which we call a cluster pseudo-marginal. We can therefore compute the difference between a state marginal and its associated polytope pseudo-marginal:

**Theorem 2.** *Let $\mathcal{C}$ be a partitioning of the state space $S$. Let $\rho \in \Lambda$ be a state-marginal and define its associated cluster-marginal as $\varrho(s_i \mid \mathcal{C}) = \frac{1}{|C_i|} \sum_{x_i \in C_i} \rho(x_i)$ for $s_i \in C_i$. Then, the Wasserstein distance $W$ between marginals $\rho$ and $\varrho$ under metric $\delta$ is bounded as follows:*

$$W_\delta(\rho, \varrho) \leq \sum_i \frac{1}{|C_i|} \sum_{s_i, x_i \in C_i} \delta(s_i, x_i). \tag{13}$$

We again recognize a partitioning problem on the right hand side; hence a clustering minimizes the distance between state-marginals and cluster state-marginals. In other words, **considering cluster visitations instead of state visitations when building the CCS will lead to a skill set closer and closer to optimal as the number of clusters and their quality increase**. Note that this theorem extends Proposition 1.

*Proof.* We will compute the Wasserstein distance between a random marginal $\rho$ and its discrete approximation over the clustering, $\varrho$. The Wassertstein distance (Villani et al., 2009) between distributions $P$ and $Q$, defined in a metric space $(\mathcal{X}, \delta)$, corresponds to the minimum cost of transporting P into Q:

$$W_\delta(P, Q) \doteq \inf_{\lambda \in \Lambda(P,Q)} \int_{\mathcal{X} \times \mathcal{X}} \delta(x, y) \lambda(x, y) \, dx dy$$

where $\Lambda(P, Q)$ is the set of all couplings of $P$ and $Q$. The Monge-Kantorovich duality shows that the Wasserstein has a dual form (Müller, 1997):

$$W_\delta(P, Q) = \sup_{f \in 1\text{-Lip}} |\mathbb{E}_{x \sim P} f(x) - \mathbb{E}_{y \sim Q} f(y)|$$

where 1-Lip is the set of 1-Lipschitz functions under metric $\delta$, $|f(x) - f(y)| \leq \delta(x, y)$. Under this dual form, we can compute the difference between the true state marginal $\rho$ and its discrete approximation $\varrho$:

$$W\left(\rho,\varrho\right)=\sup_{f:1\text{-Lip}}\left|\sum_{s\in\mathcal{S}}\left[\rho\left(s\right)-\varrho\left(s\mid\varsigma\right)\right]f\left(s\right)\right|$$

$$=\sup_{f:1\text{-Lip}}\left|\sum_{i}\sum_{s_i\in C_i}\rho\left(s_i\right)f\left(s_i\right)-\sum_{i}\sum_{s_i\in C_i}\frac{1}{|C_i|}\sum_{x_i\in C_i}\rho\left(x_i\right)f\left(s_i\right)\right|$$

$$=\sup_{f:1\text{-Lip}}\left|\sum_{i}\sum_{s_i\in C_i}\rho\left(s_i\right)f\left(s_i\right)-\sum_{i}\sum_{x_i\in C_i}\rho\left(x_i\right)\underbrace{\frac{1}{|C_i|}\sum_{s_i\in C_i}f\left(s_i\right)}_{\doteq f_i}\right|$$

$$\leq\sup_{f:1\text{-Lip}}\sum_{i}\sum_{s_i\in C_i}\left|f\left(s_i\right)-f_i\right|$$

Since $f$ is 1-Lipschitz, i.e., $\left|f\left(s_1\right)-f\left(s_2\right)\right|\leq\delta\left(s_1,s_2\right)$:

$$\left|f\left(s_i\right)-f_i\right|=\left|f\left(s_i\right)-\frac{1}{|C_i|}\sum_{x_i\in C_i}f\left(x_i\right)\right|$$

$$\leq\frac{1}{|C_i|}\sum_{x_i}\left|f\left(s_i\right)-f\left(x_i\right)\right|$$

$$\leq\frac{1}{|C_i|}\sum_{x_i}\delta\left(s_i,x_i\right)$$

Therefore, finally

$$W_\delta\left(\rho,\varrho\right)\leq\sum_{i}\frac{1}{|C_i|}\sum_{s_i,x_i\in C_i}\delta\left(s_i,x_i\right).$$

$\square$

## B  Future Work

**Learning USFAs.** The challenge of accurately learning a USFA $\psi\left(s,a;\mathbf{w}\right)$ is arguably more complex than that of learning the features $\phi$—indeed, the network needs to be able to learn and generalize optimal policies over *any* reward function encoded via a weight vector $\mathbf{w}$. As a result, the state-of-the-art performance reported in the URL Benchmark (Laskin et al., 2021) for USFA-based agents is still far from the performance of agents trained to be specialized in each task separately. First, we note that selecting the task $\mathbf{w}$ that an agent will train next can be seen as a form of curriculum learning. In the benchmark we employed in our experiments, the task vectors $\mathbf{w}$ the agent trains on are uniformly randomly sampled. Instead, we can employ existing methods to identify the best next task $\mathbf{w}$ to train an agent on (Alver & Precup, 2022b; Alegre et al., 2022) to increase the sample efficiency of our method. Second, in the URBL benchmark, agents directly follow and learn from the greedy policy $\pi\left(a\mid s;\mathbf{w}\right)\approx\arg\max_a\psi\left(s,a;\mathbf{w}\right)\cdot\mathbf{w}$. Instead, we can exploit General Policy Improvement (Barreto et al., 2017a, GPI) and re-use previously learned weight vectors when solving a new task $\mathbf{w}$ (Borsa et al., 2019), in both training and testing phases. Third, there is room for improving the capability of USFAs to generalize across different tasks $\mathbf{w}$. Current methods may not scale to practical applications with large high-dimensional task spaces. To address this, we note that often multiple similar weight vectors $\mathbf{w}$ lead to exactly the same policies (e.g., when adding more positive weights in rewards along the optimal trajectory). If future work could foster these symmetries, USFA learning could be significantly accelerated.

**Hierarchical RL.** SFs have also been used to build a strong and adaptive controller for Hierarchical RL (Machado et al., 2023). In this context, the agent drives the online exploration of the environment while building a behavior basis through SFs. For instance, in the Option Keyboard (Barreto et al., 2019) framework, a hierarchical meta-policy can be learned to output task vectors, $\pi(\mathbf{w} \mid s)$, which are given as input to the low-level policy in order to trigger precise behaviors. Such a meta-agent can, therefore, abstract out the fine-grained controls to focus on higher-level temporal reasoning. We believe these ideas strongly align with SCs, as the reasoning of the meta-agent can be visualized through the sequence of task vectors $\mathbf{w}$ it produces. We also note that there is already a natural hierarchical structure in the SCs framework presented in this paper. Indeed, we have observed that the USFA $\boldsymbol{\psi}(s, a; \mathbf{w})$ can induce policies that traverse all state clusters but cannot fine-tune their position within a cluster. However, we have also trained a goal-reaching policy via $Q(s, a; g)$, able to navigate at shorter time-frames. It is, therefore, straightforward to activate the goal-reaching policy to the highest-rewarding state $g = \arg\max_{s \in C_i} r(s)$ once the highest-valued cluster $C_i$ is reached. This enables the agent to perform within-cluster movement and addresses one of the most prominent limitations of our method. We demonstrate this in Appendix C.6. It might also be possible to activate the goal-reaching policy at different times during the trajectory, we leave the combination of SCs and hierarchical techniques as a promising future research direction.

**Clustering methods.** In the previous section, we discussed the problem of the sampling distribution induced by the replay buffer in an offline RL setting. This leads to a discrepancy between Theorem 1, formulated as a sum/integral over the state space, and the sampling distribution we have access to. Algorithms like K-Means can possibly over-sample regions following the sampling distribution. We argue that a better clustering objective is to minimize the radius of the cluster, to ensure all regions cover around the same amount of time-steps. This objective is sometimes called Metric K-Centers (Hakimi, 1964) and has good approximate solutions. However, this objective itself can be susceptible to outliers and hard to compute online; further study on clustering algorithms is required. With the right approach, a non-linear clustering algorithm like DBSCAN could be a promising alternative. In our experiments, we use fixed clusters of the farthest-first traversal algorithm (K-Centers) in the Maze domain and an Online K-Means algorithm for the robotics domains for the sake of online performance comparison.

**Model-based RL and SCs.** As briefly mentioned in Section 1, model-based RL (MBRL) could, in theory, solve zero-shot RL provided a perfect model and infinite search depth. This is, of course, unrealistic, as model errors compound for deeper searches and longer horizons. However, the potential of MBRL for zero-shot RL remains, especially for SCs. As we have discussed, SCs allow to perform large-scale movement in the MDP, navigating through potentially hundreds of regions. However, they are unable to fine-tune their position within the clusters. In contrast, model-based methods excel at short-term planning and optimization before compounding errors become too big of an issue. We therefore believe they could complement each other greatly, where the long-term SC approximation can be used for bootstrapping at the leaves of a search of depth $k$. Several frameworks have been proposed for combining SFs, learning models, and planning (Bagot et al., 2023; Alegre et al., 2023).

We believe all of the future research ideas discussed above could yield significant improvements to the field of zero-shot RL, which is critical to moving toward truly general agents.

## C Experiments Details

### C.1 Hyperparameters

The hyperparameter used in this paper were mostly taken from Touati et al. (2023). Table 1 summarizes them, with a section for hyperparameters introduced in this paper.

The architecture and training algorithms (TD3) for the USFA and goal-conditioned agent were also taken from Touati et al. (2023). $z$ in the following quote refers to $\mathbf{w} \in \mathbb{R}^d$ for the USFA and $g \in \mathcal{S}$ for the goal-conditioned agent: "we first pre-process $(s, a)$ and $(s, z)$ separately by two feedforward networks with two hidden layers (each with 1024 units) to 512-dimensional space. Then we concatenate their two outputs and

Table 1: Hyperparameters used in the Experiments (Section 5).

| Hyperparameters from Touati et al. (2023) | Value |
| --- | --- |
| Replay buffer size | $5 \times 10^6$ ($10 \times 10^6$ for maze) |
| Representation dimension $d$ | 50 (100 for maze) |
| Batch size | 1024 |
| Discount factor $\gamma$ | 0.98 (0.99 for maze) |
| Optimizer | Adam |
| Learning rate | $10^{-4}$ |
| Mixing ratio for $\mathbf{w}$ sampling | 0.5 |
| Momentum coefficient for target networks | 0.99 |
| Stddev $\sigma$ for policy smoothing | 0.2 |
| Truncation level for policy smoothing | 0.3 |
| Number of gradient steps | $10^6$ |
| Number of reward labels for task inference | $10^4$ |
| Regularization weight for orthonormality loss (Laplacian) | 1 |

| Hyperparameters from this paper | Value |
| --- | --- |
| Latent space dimension $n$ | 50 |
| Non-stationary K-Means learning rate | 0.1 |

pass it into another 2-layer feedforward network (each with 1024 units) to output a $d$-dimensional[6] vector". The policy network is the same, with the output space being the action space, applying a `Tanh` activation since the actions are bounded in $[-1, 1]$.

A goal-conditioned agent was used in Touati et al. (2023) as well for the sake of zero-shot comparison. Similarly to their approach, we use the `mix_ratio` hyperparameter to sample 50% of the goals as the next state in the transition and 50% as other random states in the batch (following Hindsight Experience Replay ideas). In order to accelerate the learning of the goal-conditioned agent, we used a pessimistic initialization at $-1/(1 - \gamma)$ to avoid longer convergence times due to the overestimation bias.

### C.2 Reward Functions used in the Experiments (DeepMind Control Suite)

In Figure 14, we include descriptions of the environments and reward functions introduced by Touati et al. (2023), which were used in the experiments presented in Section 5.1.

### C.3 Comparison with the Forward-Backward Representation Method

In Figure 15, we extend the results shown in Figure 5 by including the Forward-Backward (FB) representation method (Touati et al., 2023). Although our primary goal is to design reward features that improve upon Successor Feature-based techniques, our method also achieves competitive results—outperforming FB in two out of three robotic tasks. FB is a state-of-the-art technique for unsupervised zero-shot reinforcement learning that does not fall within the class of methods we aim to improve upon, as discussed at the end of Section 5.1.

### C.4 Order for Learning Cluster Features

When learning the goal-reacher and clusters, it is most intuitive to train all elements in order—first $Q(s, a; g)$, then $\varphi(s)$, and finally $\{C_i\}_{i=1}^d$ to generate $\boldsymbol{\phi} = [\mathbb{1}_{C_1}(s), \cdots, \mathbb{1}_{C_d}(s)]$. However, the benchmarking study of Touati et al. (2023) trains the reward features within a single loop. For fairness of comparison, we also train

---

[6]Replace by the state space dimension instead of $d$ for the goal-conditioned agent.

- **Point-mass Maze**: a 2-dimensional continuous maze with four rooms. The states are 4-dimensional vectors consisting of positions and velocities of the point mass $(x, y, v_x, x_y)$, and the actions are 2-dimensional vectors. At test, we assess the performance of the agents on 20 goal-reaching tasks (5 goals in each room described by their $(x, y)$ coordinates).
- **Walker**: a planar walker. States are 24-dimensional vectors consisting of positions and velocities of robot joints, and actions are 6-dimensional vectors. We consider 4 different tasks at test time: `walker_stand` reward is a combination of terms encouraging an upright torso and some minimal torso height, while `walker_walk` and `walker_run` rewards include a component encouraging some minimal forward velocity. `walker_flip` reward includes a component encouraging some minimal angular momentum.
- **Cheetah**: a running planar biped. States are 17-dimensional vectors consisting of positions and velocities of robot joints, and actions are 6-dimensional vectors. We consider 4 different tasks at test time: `cheetah_walk` and `cheetah_run` rewards are linearly proportional to the forward velocity up to some desired values: 2 m/s for `walk` and 10 m/s for `run`. Similarly, `walker_walk_backward` and `walker_run_backward` rewards encourage reaching some minimal backward velocities.
- **Quadruped**: a four-leg ant navigating in 3D space. States and actions are 78-dimensional and 12-dimensional vectors, respectively. We consider 4 tasks at test time: `quadruped_stand` reward encourages an upright torso. `quadruped_walk` and `quadruped_run` include a term encouraging some minimal torso velocities. `quadruped_walk` includes a term encouraging some minimal height of the center of mass.

Figure 14: Description of the environments as provided by Touati et al. (2023), based on the *DeepMind Control Suite* (Tassa et al., 2018).

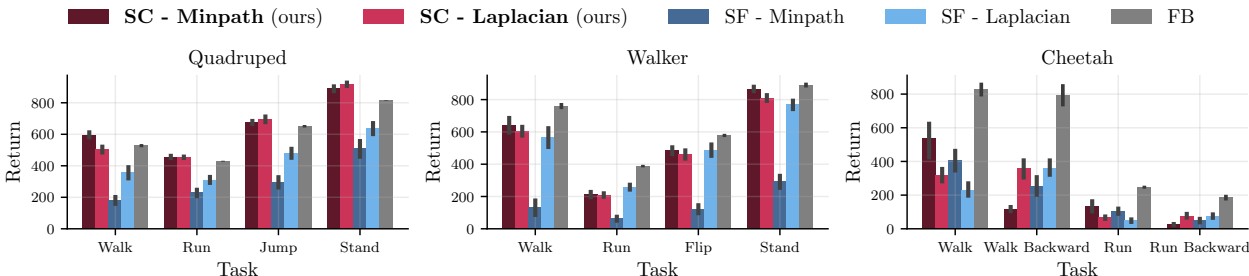

Figure 15: Mean returns obtained in the Quadruped, Walker, and Cheetah tasks from the URLB benchmark. We compare two variants of our method, Successor Clusters (SC), shown in shades of red, and two Successor Feature (SF)-based techniques, shown in shades of blue. Both sets of variants are based on Minpath and Laplacian reward features. The Forward-Backward (FB) method is shown in grey.

all elements simultaneously in the quantitative experiments. We use an online and non-stationary version of K-Means that updates the cluster centers with exponential smoothing with weight 0.1.

### C.5   Reward Families Employed in Section 5.2

In Figure 16, we display 5 instances of reward functions from each of the reward families generated for the experiments discussed in Section 5.2.

### C.6   Hierarchically Combining Successor Clusters and Goal-Conditioned RL

While this is beyond the scope of the zero-shot representation we aim to build in this paper, in Section B we discuss the possibility to combine the smooth, long-term movement of our SCs USFA agent with the more precise and short-term movement of the goal-conditioned policy. This can be achieved quite easily by using the SCs USFA agent until the highest valued cluster is reached, $s \in C_j \mid j = \arg\max_i w_i$. We can then switch to the goal-reaching policy: its goal can then simply be the highest-valued state within

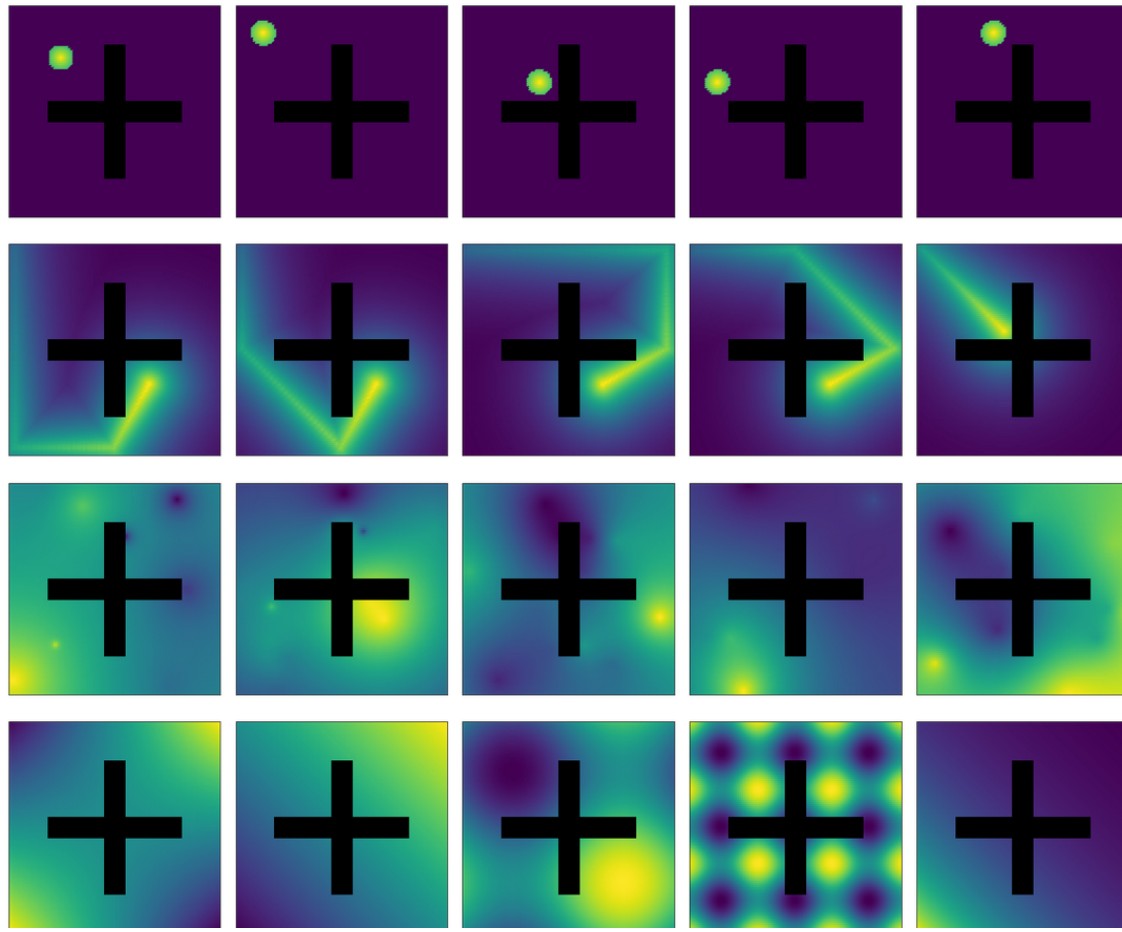

Figure 16: The 4 reward families chosen for reward prediction evaluation. By row, top to bottom: goal-reaching, trajectories, Gaussians, smooth maps.

the cluster, $g = \arg\max_{s \in C_j} r(s)$. This will fine-tune the position of the agent within the cluster, though it will ignore local reward structure to only reach the highest-value state; arguably this is a reasonable approximation to make locally. We demonstrate this simple improvement on Successor Clusters in Figure 17, on the same reward function as presented in the experiments in Figure 11b. We can see that the resulting trajectory reaches the highest-value state in the cluster, while the SCs USFA agent stopped in the border of the highest-valued cluster.

## D    Code snippets

In this section, we provide simplified code snippets for Successor Clusters: the clustering and minimal path function. This code assumes usage of the URLB repository (Laskin et al., 2021) with the SFs implementation from Touati et al. (2023).

First, we provide a clustering wrapper that can be used on top of any **FeatureLearner**, using it as a latent space learner to provide cluster occupancy reward features (file **sf.py**). In the context of this code, this requires a non-stationary K-Means implementation:

```
class NonStat_KMeans:
    def __init__(self, n_clusters, alpha=0.1):
        self.n_clusters = n_clusters
        self.cluster_centers_ = None
        self.alpha = alpha
```

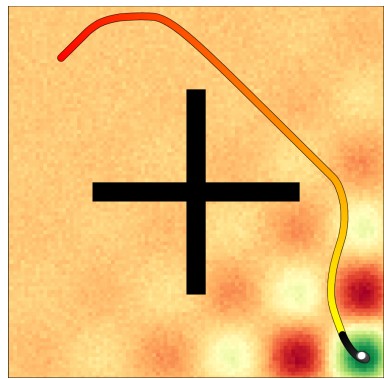

Figure 17: Using the SCs USFA (trajectory red-to-yellow) until the highest-reward cluster is reached, from which point we trigger the goal-reaching policy to reach the highest-reward state in the cluster (trajectory black-to-white). The background corresponds to the true reward function, as in Figure 11b.

```python
6
7    def partial_fit(self, data):
8        N, dim = data.shape
9        alpha = self.alpha
10       # initialize centers (naive: random elements)
11       if self.cluster_centers_ is None:
12           self.cluster_centers_ = data[torch.randperm(N)[:self.n_clusters]].clone()
13           alpha = 1 # first step: directly update to the new centers
14       # Lloyd's algorithm (assign & update)
15       dists = torch.cdist(data, self.cluster_centers_)
16       _,i = dists.min(-1)
17       new_cluster_centers_ = torch.stack([data[i==j].mean(0) for j in range(self.n_clusters)])
18       self.cluster_centers_ = self.cluster_centers_*(1-alpha) + alpha*new_cluster_centers_
19
20   def predict(self, data):
21       dists = torch.cdist(data, self.cluster_centers_)
22       _,i = dists.min(-1)
23       return i.long()
```

Listing 1: Simple Non-stationary K-Means.

We can use this to build the cluster occupancy reward features:

```python
1    class ClusterLearner(FeatureLearner):
2        """ FeatureLearner wrapped around a FeatureLearner (that is used for latent space).
3        Breaks it down into clusters and uses phi=[s in cluster 1, ....] """
4        def __init__(self, feature_learner, z_dim, device) -> None:
5            self.feature_learner = feature_learner
6            self.z_dim = z_dim
7            self.device = device
8            self.cluster_algo = NonStat_KMeans(n_clusters=z_dim)
9            self.fitted = False
10
11       def forward(self, obs: torch.Tensor, action: torch.Tensor, next_obs: torch.Tensor,
    future_obs: torch.Tensor):
12           phis = self.feature_learner.feature_net(obs).detach()
13           self.cluster_algo.partial_fit(phis)
14           self.feature_learner(obs, action, next_obs, future_obs)
15
16       def feature_fc(self, obs):
17           phis = self.feature_learner.feature_net(obs).detach()
18           onehot_clusters = F.one_hot(self.cluster_algo.predict(phis), num_classes=self.z_dim)
19           return onehot_clusters.to(self.device).to(torch.float)
```

Listing 2: Cluster Occupancy Reward Features.

Our implementation is based on the minimum distance between two states. Assuming a provided goal-conditioned agent (`GoalReacher`) that can estimate the $Q$-value in Equation (8):

```python
class MinimalPath(FeatureLearner):
    """ Assuming a trained GoalReacher agent,
    the features approximate the average minimal path from s to g. """
    def __init__(self, obs_dim, action_dim, z_dim, hidden_dim) -> None:
        super().__init__(obs_dim, action_dim, z_dim, hidden_dim)
        self.goalreacher = GoalReacher(obs_dim, action_dim, z_dim, hidden_dim)

    def forward(self, obs: torch.Tensor, action: torch.Tensor, next_obs: torch.Tensor,
    future_obs: torch.Tensor):
        goals = next_obs[torch.randperm(len(next_obs))]
        # Phi loss
        latent_obs = self.feature_net(obs)
        latent_goals = self.feature_net(goals)
        dist_pred = (latent_obs - latent_goals).pow(2).sum(-1)
        dist_pred = torch.sqrt(torch.clamp(dist_pred, min=1e-6))
        dist_q = self.goalreacher.compute_distance(obs, goals)
        latent_loss = (dist_q - dist_pred).pow(2).mean()
        return latent_loss
```

Listing 3: Learning a Minpath Latent Space from a GoalReacher Agent.

The `GoalReacher` agent can either be trained in tandem with the latent space or at an earlier stage; it can be implemented in many ways, for example offline with the `update` function below:

```python
class GoalReacher:
    """ A Q value learns the shortest path from s to g, in terms of (neg) steps.
    The actor learns the associated policy (with TD3). """
    def __init__(self, obs_dim, action_dim, z_dim, hidden_dim) -> None:
        super().__init__(obs_dim, action_dim, z_dim, hidden_dim)
        self.qnet = ForwardMap(obs_dim, obs_dim, action_dim,
                               hidden_dim//2, hidden_dim, out_dim=1,
                               preprocess=True, add_trunk=False) # same as USFA
        self.qnet_targ = ForwardMap(obs_dim, obs_dim, action_dim,
                                    hidden_dim//2, hidden_dim, out_dim=1,
                                    preprocess=True, add_trunk=False) # same as USFA
        self.actor = Actor(obs_dim, obs_dim, action_dim,
                           hidden_dim//2, hidden_dim,
                           preprocess=True, add_trunk=False) # same as USFA

        lr = 1e-4
        self.actor_opt = torch.optim.Adam(self.actor.parameters(), lr=lr)
        self.q_opt = torch.optim.Adam(self.qnet.parameters(), lr=lr)

        self.apply(utils.weight_init)

    def update(self, obs: torch.Tensor, action: torch.Tensor, next_obs: torch.Tensor):
        actor_loss, q_loss = self.compute_losses(obs, action, next_obs)
        # Optimize actor
        self.actor_opt.zero_grad(set_to_none=True)
        actor_loss.backward()
        self.actor_opt.step()
        # Optimize Q & update target
        self.q_opt.zero_grad(set_to_none=True)
        q_loss.backward()
        self.q_opt.step()
        utils.soft_update_params(self.qnet, self.qnet_targ, 0.01)

    def compute_losses(self, obs: torch.Tensor, action: torch.Tensor, next_obs: torch.Tensor
    , future_obs: torch.Tensor):
        # Hindsight Experience Replay: first 50% of the transitions are successful
        mid = 1024//2 # mix_ratio = 0.5
        shuffle = torch.stack([torch.arange(mid),torch.randperm(mid)+mid]).flatten()
        goals = next_obs[shuffle]
```

```
40
41          # Actor actions
42          best_action = self.actor(obs, goals, .2).sample(clip=.3)
43          best_next_action = self.actor(next_obs, goals, .2).sample(clip=.3).detach()
44
45          # TD3 for Q function learning
46          qsg1, qsg2 = self.qnet(obs, goals, action)
47          next_qsg1, next_qsg2 = self.qnet_targ(next_obs, goals, best_next_action)
48          next_qsg = torch.min(next_qsg1.squeeze(), next_qsg2.squeeze()).detach()
49          done = torch.all(next_obs==goals, dim=-1).float().detach()
50          q_bell = -1 + 0.99 * next_qsg * (1-done)
51          q_loss1 = (qsg1.squeeze() - q_bell.detach()).pow(2).mean()
52          q_loss2 = (qsg2.squeeze() - q_bell.detach()).pow(2).mean()
53
54          # Actor loss
55          best_qsg1, best_qsg2 = self.qnet(obs, goals, best_action)
56          best_qsg = torch.min(best_qsg1.squeeze(), best_qsg2.squeeze())
57          actor_loss = -best_qsg.mean()
58
59          return actor_loss, q_loss1 + q_loss2
60
61      def compute_distance(self, obs, goals):
62          best_action_sg = self.actor(obs, goals, .2).sample(clip=.3)
63          dist_q_sg = self.qnet(obs, goals, best_action_sg).detach()
64          best_action_gs = self.actor(goals, obs, .2).sample(clip=.3)
65          dist_q_gs = self.qnet(goals, obs, best_action_gs).detach()
66          return -(dist_q_sg + dist_q_gs)/2
```

Listing 4: Training a GoalReacher Agent.

