# OpenReview forum: "Successor Clusters: A Behavior Basis for Unsupervised Zero-Shot Reinforcement Learning"
_TMLR — Accepted by TMLR_

### Review · Reviewer_Jgad · 2025-03-21

**Summary Of Contributions:**

This paper introduces Successor Clusters, a new method for unsupervised zero-shot reinforcement learning. It tries to enable an RL agent to adapt to new reward functions without any additional training. This is achieved by clustering the state space using a novel distance metric based on the minimum number of time steps to transition between states. SCs, a variant of Successor Features, predict the expected time a policy spends in each of these state clusters. The paper demonstrates that Successor Clusters can approximate and maximize new reward functions in a zero-shot manner and shows that the method produces interpretable features, allowing visualization of an agent's likely trajectory. Empirical results show that SCs outperform state-of-the-art methods in continuous control benchmarks, achieving better zero-shot performance and lower reward approximation error.

**Audience:**

Yes

**Claims And Evidence:**

Yes

**Requested Changes:**

Overall, I think this paper is in good shape.

**Strengths And Weaknesses:**

## Strengths

- SCs enables agents to adapt to new reward functions without any additional training. The paper empirically demonstrates that SCs outperform state-of-the-art methods in continuous control tasks..
- The paper provides a strong theoretical foundation for SCs. It introduces a mathematical formulation for learning reward features and proves that SCs allow for arbitrarily good approximation of any Lipschitz reward function. It also shows that as the number and quality of clusters increase, the set of policies induced by SCs converges to a set containing the optimal policy for any new task.
- SCs offer a high degree of interpretability. Because SCs predict the expected time an agent spends in each state cluster, it's possible to visualize the agent's planned trajectory and understand its behavior. This contrasts with other methods that often rely on less interpretable features.
- The method allows for controlling the agent's behavior by adjusting the weights associated with different clusters. It also allows for specifying desired trajectories in terms of cluster visits, enabling a form of imitation learning.



## Weaknesses

- The specific clustering algorithm used (K-Means in the experiments) has its own limitations, such as sensitivity to initialization and the assumption of isotropic clusters. While the paper mentions exploring other clustering algorithms (like DBSCAN or Metric K-Centers) as future work, the current implementation relies on K-Means or its variants.

- The Universal Successor Feature Approximator needs to learn and generalize optimal policies over any reward function, which might be a more complex task than learning the features themselves. This can limit the scalability of the method.

---

> ### Author Response · Authors · 2025-06-02
>
> We thank the reviewer for the constructive feedback. We are glad that the reviewer found our method to be well-motivated, theoretically grounded, and empirically effective. Please find below our responses to the reviewer’s suggestions.
>
> ---
>
> ### Suggested future work: Exploring alternative clustering algorithms
>
> We agree that K-Means may present limitations in certain settings, especially due to its sensitivity to centroid initialization and its tendency to discover spherical, similarly-sized clusters. In this paper, we investigated the performance of our method using K-Means as the underlying clustering method due to its simplicity, interpretability, and strong performance in online settings. This made it a suitable starting point to demonstrate the feasibility and effectiveness of Successor Clusters. That said, since our framework is agnostic to the specific clustering algorithm used, we agree with the reviewer that exploring alternative clustering methods—such as DBSCAN or Metric K-Centers—is a promising direction for future work. Such experiments could reveal, for example, to what extent more sophisticated clustering techniques might further improve the strong zero-shot performance already achieved by our method.
>
> ---
>
> ### Learning Universal Successor Feature Approximators (USFAs)
>
> This is a valuable observation. Indeed, we view the challenges of training universal approximators as an important orthogonal challenge in unsupervised zero-shot RL. USFAs have been extensively studied in the literature (e.g., Barreto et al., 2019; Touati et al., 2023) and have been shown to perform well in a variety of domains. Our experiments further confirm that, when combined with a structured feature representation like Successor Clusters, USFAs remain effective at generalizing in multi-task settings.
>
> That said, we do agree that the scalability of USFA-like approaches deserves further investigation. We are currently working on techniques to improve how USFAs are trained and optimized, and we look forward to continued progress from the community—as new methods for learning USFAs emerge, they can be readily and directly integrated into our framework.
>
> ---
>
> Once again, we thank the reviewer for their thoughtful comments and suggestions! We appreciate the constructive ideas for future work and would be happy to continue the conversation should any additional points arise.

---

### Review · Reviewer_67KV · 2025-03-22

**Summary Of Contributions:**

1. The paper proposes a reward coverage maximization problem in the successor feature (SF) learning scheme, which tries to select feature that well-approximates the reward function. The error is upper bounded by some distance measure, and the authors proposes a method that minimizes this upper bound motivated from the SPP.

2. The paper considers a $K$-lipschitz reward functions with respect to the state, guaranteeing smoothness of reward functions as the state changes.

3. The paper considers occupancy measure in terms of the clusters.

4. A distance function is learned to identify the clusters that minimizes the error bound in (4).

**Audience:**

Yes

**Claims And Evidence:**

Yes

**Requested Changes:**

1. The class of reward function requires more context around it. For example Tasfi et al. considers quadratic form of rewards for SF learning.

2. The contribution part can be written in a more concise manner. For example, the last item seems to be merged into the fourth item.


3. The reward function seems to be deterministic from each state. Several comments is required when the reward function depends also on the action, and its next state.

Tasfi, Norman L., and Miriam Capretz. "Non-Linear Rewards For Successor Features." (2020).

**Strengths And Weaknesses:**

**Strength**
1. The literature in SF often assume the reward function is linear in terms of successor feature. This paper broadens this concept to lipschitz rewards.

2. The concept of successor cluster is reasonable and a simple and clear idea. The authors explore this idea, and provides interesting properties of this concept.

3. The paper is well-written and easy to follow.


**Weakness**

1. Proposition 1 can be improved : How many clusters do we need or what is the dependency on the problem parameters to cover an optimal policy?

2. It is not clear how tight the error bound on (4) is.

---

> ### Author Response · Authors · 2025-06-02
>
> We thank the reviewer for carefully reading our paper and for the positive comments and constructive suggestions to improve the clarity of our contributions.
>
> Below, we respond to each of the reviewer’s comments and address all requested changes.
>
> ### Requested Change 1: Additional motivation for the class of reward functions
>
> We thank the reviewer for raising this important point. As the reviewer correctly noted, alternative classes of reward functions—such as quadratic—have been considered in the context of zero-shot transfer. That said, recent independent studies have consistently shown that methods leveraging linear-reward-function approximation (and the Successor Feature framework, SF) can support state-of-the-art performance across a wide range of challenging benchmarks. See, for example, the thorough studies conducted by Touati et al. (2023), where methods leveraging linear reward models and SFs were capable of producing state-of-the-art zero-shot performance across several domains. Among these, one particular family of approaches—Laplacian-based methods—was reported as *“the only [ones] that perform[ed] consistently well, both over tasks and over replay buffers”*. This observation emerged from a careful comparison involving ten baselines based on linear approximators and Successor Features (SFs), as well as a Forward-Backward (FB) approach that also targeted zero-shot transfer. FB does not leverage linearly decomposed reward functions, placing it in a broader function class than ours—but at the cost of significantly higher computational complexity.
>
> In addition to their strong empirical performance, linear reward formulations—particularly those integrated into the Successor Features framework—have received sustained and growing attention (see, e.g., the observations and findings by Carvalho et al. 2023a,b and Touati et al. 2023). Prominent examples of such formulations are those based on linear eigenfunctions (Wu et al., 2019), which form a Fourier-like basis for approximating value functions in an MDP and are widely regarded among the current state-of-the-art methods for zero-shot transfer.
>
> Given the well-established potential of this class of reward functions to support strong zero-shot learning performance, and the sustained attention it has received in recent work, we chose to investigate it by introducing a novel and principled method within this particular class of techniques. Specifically, our goal was to develop an approach that builds on and extends the Successor Features framework and that is capable of consistently outperforming existing zero-shot transfer methods based on the linear-reward assumption.
>
> We appreciate the reviewer pointing out that the class of linear-reward methods could be more clearly situated within the broader zero-shot transfer literature. We will update the manuscript accordingly to further clarify the motivations underlying our focus on this class.

---

> > ### Author Response · Authors · 2025-06-02
> >
> > ### Requested Change 2: More concise description of contributions
> >
> > Thank you for this suggestion. We have followed the reviewer’s recommendation and merged the fourth and sixth bullet points in the Introduction, which has indeed made the description of our contributions more concise and easier to follow.
> >
> > ---
> >
> > ### Requested Change 3: Discussion of state-action and stochastic reward functions
> >
> > These are excellent points—thank you for bringing them up. As stated in the footnote on page 4, to simplify notation, we focus on the case of state-based reward functions $r(s′) = r(s,a,s′)$. Importantly, however, all definitions can be easily extended to the case of more general reward functions—including $r(s,a)$ and $r(s,a,s’)$. Regarding reward stochasticity, please note that this does not affect our analyses or algorithm: since RL methods optimize expected returns, any stochastic reward function can, without loss of generality, be rewritten as a deterministic one by replacing it with its expected value. This preserves all relevant properties of the MDP and does not affect the applicability of our framework. We will update the manuscript accordingly to explicitly discuss these points and clarify why these notational simplifications do not affect the proposed technique or the validity of our formal results.
> >
> > ---
> >
> > ### Does Proposition 1 provide insight into how many clusters may be needed to cover optimal policies?
> >
> > This is an excellent point. Proposition 1 (now renamed Observation 1) is a counting argument and is not meant to provide convergence guarantees. That said, Theorem 2 in Appendix A.2 offers exactly the type of analysis the reviewer is referring to. Specifically, it builds on the concept of state marginal polytopes introduced by Eysenbach et al. (2022) to derive bounds—similar in form to those of Theorem 1—that describe conditions under which the policy set is guaranteed to become optimal, depending on the number of clusters. We will add a brief discussion following Observation 1 to highlight this connection and refer the reader to Appendix A.2 for further details on how the number of clusters relates to optimality.
> >
> > ---
> >
> > We hope our responses address the main points you raised. If any points remain unclear, we welcome further feedback and would be happy to continue the discussion. Thank you again for your thoughtful and constructive comments and suggestions!
> >
> > ---
> >
> > Eysenbach, Benjamin, Salakhutdinov, Ruslan, and Levine, Sergey. “The Information Geometry of Unsupervised Reinforcement Learning”. ICLR, 2022.

---

### Review · Reviewer_Mwrx · 2025-05-19

**Summary Of Contributions:**

This paper addresses the problem of zero-shot unsupervised RL, where the environment $(\mathcal{S},\mathcal{A},\gamma,p,\mu)$ keeps constant, but the reward function can be arbitrary, and is revealed only after a reward-free exploration with the environment. To address it, the authors adopt the method of successor features existing in the literature, that is based on learning the successor features $\psi(s,a;w)$ for a subset of policies "linear" in $w$, for any $w$.

In this context, the main contribution of the authors is proposing a new method for finding the features $\phi$. Specifically, they propose to clusterize the state space using as distance between states $s,s'$ the minimum expected number of stages for going from $s$ to $s'$ (the average of $s\to s'$ and $s'\to s$), and then to use the one-hot features $\phi(s)$ representing the membership to of state $s$ to each cluster $\\{C_i\\}_i$.

Finally, the authors present a practical algorithm for unsupervised zero-shot transfer in RL and conduct some experimental validation.

**Audience:**

Yes

**Broader Impact Concerns:**

/

**Claims And Evidence:**

No

**Requested Changes:**

Please, see the "Strengths And Weaknesses" section.

In particular, the following adjustments are critical:

-  Answer to all my questions :)
- Avoid defining a uniform prior over an unbounded set (it is not possible)
- Fix Theorem 1 (if possible)
- Improve a lot the presentation of Section 4.
- Conduct experiments with other baselines, specifically with forward-backward representations as in Touati et al. 2023.

One of my biggest concerns is with the last point. Touati et al. 2023 explain that forward-backward representations work quite good for the task of zero-shot unsupervised RL, so why using a successor feature method? Please, motivate.

**Strengths And Weaknesses:**

## Strengths:

- The idea of using Eq. (5), i.e., the minimum expected number of stages between different states, for constructing features for the state space is interesting.

## Weaknesses:

Overall, I find the contributions of the paper quite limited, as "successor clusters" means just proposing a new heuristic approach for computing the features of the state space, and this heuristic approach does not convince me very much, because (1) there are some problems in the theoretical results, and (2) the experiments compare do not compare with enough baselines.
In particular:

- In Fig. 1 symbols $Q(s,a;g)$ and $\psi(s,a;w)$ are undefined. I suggest to provide in the caption a note to explain their meaning.
- At beginning of Section 3, symbol $\mathcal{W}$ is undefined.
- In Section 3.1, you say that: "The main obstacle to solving this problem is that we do not have prior information on the reward functions that an agent may need to solve during transfer. Hence, the representation $\phi$ of dimension $d$ must linearly cover the widest range of possible reward functions uniformly sampled from the given space of reward functions". Why is this? I mean, this objective is equivalent to assuming a "uniform prior" over rewards in $\Gamma$. However, note that, since $\Gamma$ is unbounded, then there is no notion of "uniform prior" that can be defined onto it. Simply put, my guess is that if $d<S$, then for whatever choice of $\phi$ you make, the objective in Eq. (3) is $+\infty$.
- What sense does it have to upper bound $1/|C_i|^2$ with $1/|C_i|$ in the proof of theorem 1?
- In the proof of theorem 1, I think that the passage associated to (triangle ineq.) if wrong. Basically, you are using that $(\sum_i a_i)^2\le \sum_i a_i^2$, but how can you say that?
- In addition, Theorem 1 is wrong because set $\Gamma_{K-Lip}$ is unbounded, and so we cannot define a uniform distribution on it.
- In Fig. 3 there is a error/typo: $\gamma=1$ cannot be. P.S.: I realized later that you use terminal states. Please, make it clear in the background section adding a proper formalism.
- The proof of Proposition 1 is missing. However, it is obvious so that it does not require a proof. As such, you should not call it "proposition", but use something like "fact", or "observation".
- where is $\pi$ used in Line 7 of Algorithm 1?
- In Algorithm 1, what is $Q(s,a;g)$? P.s.: found it. Please, make it clearer.
- What is $\mathcal{W}$ in Eq. (7)? I mean, how one should set it?
- You should write "Euclidean" and not "Euclidian".
- In Section 4.2, if you have $n$ samples from the reward, how can you compute Eq. (10)? I mean, you do not know $r$ everywhere.
- In the experiments, why do you not compare with forward-backward representations as in Touati et al. 2023? Why do not you try with other baselines, i.e., other kind of features (beyond those in Touati et al. 2023)?
- I cannot find in the text what reward functions did you use in the experiment Q1. Specifically, I would like to know if you sample the rewards uniformly from the set of lipschitz rewards and what is the Lipschitz constant.
- I strongly suggest to improve the presentation of section 4, because it is very difficult to understand how Algorithm 1 is implemented (some symbols/concept undefined).

---

> ### Author Response · Authors · 2025-06-02
>
> We sincerely thank the reviewer for their detailed and insightful feedback. All requested changes and clarifications have been addressed, and the corresponding revisions are highlighted in blue in the updated manuscript.
>
> Below, we address each of the reviewer’s questions and suggestions:
>
> ---
>
> ### Theorem 1 and unbounded reward space
>
> We thank the reviewer for highlighting the importance of clearly and precisely characterizing the class of possible distributions $\mathcal{P}$ over reward functions on which our objective is defined, as well as for pointing out a few (relatively minor) technical issues with the way we presented our theoretical result. We have revised the paper to address these points. Please see our detailed responses below.
>
>
> - Regarding the importance of clearly characterizing the properties of the possible distributions $\mathcal{P}$ over reward functions (Equation 3), we have updated the paper to include a focused discussion addressing this point. When the state space $S$ is finite, a reward function can be represented as a vector $r \in \mathbb{R}^{|S|}$, and $\mathcal{P}$ can be taken to be any distribution over $\mathbb{R}^{|S|}$. A natural and mathematically grounded choice—especially when little is known a priori about the structure of reward functions—is to adopt a *maximum entropy distribution*, which corresponds to the distribution that makes the fewest assumptions beyond basic constraints. For instance, if we only assume that the expected squared norm $\mathbb{E}[\|r\|^2]$ is bounded, then the maximum entropy distribution is a multivariate Gaussian with zero mean and isotropic covariance: $r \sim \mathcal{N}(0, \sigma^2 I)$. Alternatively, if we assume that reward values are bounded (e.g., $r(s) \in \left[r_{\min}, r_{\max} \right]$ for all $s$), then the maximum entropy distribution becomes the uniform distribution over the hypercube $\left[r_{\min}, r_{\max} \right]^{|S|}$. Both cases are useful in practice and provide principled ways to define expectations over reward functions while maintaining minimal prior assumptions. In more general settings, where the state space is continuous or infinite, $\mathcal{P}$ can be defined as a distribution over functions. A common choice is to use a Gaussian process (GP) prior, which generalizes the Gaussian case to infinite-dimensional function spaces. If the reward function is assumed to lie in a bounded interval, as above, then a simple and flexible prior is a GP with a constant mean function (e.g., $(r_{\min} + r_{\max})/2$) and a stationary kernel such as the squared exponential (RBF) kernel with large length-scale. This setup encourages smoothness while avoiding strong structural biases. Please note that the clarifications and comments above do not affect the correctness of Theorem 1, as the result concerns what can be viewed as a “worst-case” class of Lipschitz-continuous reward functions.
>
> - Regarding the use of the triangle inequality: We thank the reviewer for catching the oversight in our use of the triangle inequality, which stemmed from a notational simplification when transitioning from absolute to squared differences. Fortunately, this does not affect the correctness of Theorem 1 or the resulting bound. As the reviewer noted, the inequality $(\sum_i a_i)^2 \leq \sum_i a_{i}^2$ is not valid in general. However, a valid bound in this case is $(\sum_i a_i)^2 \leq n \sum_i a_{i}^2$, where $n$ is the number of terms in the sum. This can be seen from the perspective of several inequalities, such as Jensen’s inequality. In our proof, this introduces a $|C_i|$ term that cancels with the $1/|C_i|^2$ factor, resulting in the correct final scaling of $1/|C_i|$. This also addresses the reviewer’s question about the transition from $1/|C_i|^2$ to $1/|C_i|$. Importantly, as noted above, this minor correction to the proof does not affect the validity of the bound, which remains correct. We again thank the reviewer for the careful and constructive feedback.

---

> ### Author Response · Authors · 2025-06-02
>
> ### On comparing with Forward-Backward (FB) representations
>
> The reviewer is correct that FB, like our method, addresses the zero-shot learning setting. They raised the important question of whether a direct comparison would be meaningful and necessary to support our claims. Our paper tackles two core challenges:
>
> **(1) Identifying reward features that optimize the reward coverage maximization objective.**
>
> FB does not aim to identify or construct such features. Instead, it directly approximates optimal value functions $q^*$ using, among other components, learned task-dependent features $F(s,a,z)$. Importantly, in doing so, the method focuses on reconstructing the value function and does not attempt to learn features for effectively linearly approximating the reward function itself. In other words, FB focuses on learning features that allow the optimal $q$-function to be linearly reconstructed *given* a reward function. In contrast, our goal is to learn features that allow the reward functions themselves to be linearly approximated. Put differently, FB focuses on value function reconstruction, whereas our method learns features that support approximation of a broad set of reward functions. For this reason, FB is not directly comparable to our method with respect to our first key objective—reward coverage—since it does not perform reward approximation.
>
> **(2) Performing zero-shot transfer in the setting where reward functions are linearly approximated/modeled.**
>
> As stated in our paper (e.g., page 2), we focus specifically on the setting in which reward functions are assumed to be linearly approximated. This assumption naturally leads to techniques leveraging the Successor Features (SF) framework. Both the linear-reward setting and the SF framework have received considerable and sustained attention in recent years (e.g., Barreto et al. 2018; Hansen et al. 2020; Carvalho et al. 2023a,b; Touati et al. 2023). Our method was specifically developed to operate under this assumption and builds directly on (and extends) this framework. As such, we compare against techniques that operate under the same assumptions. FB does not fall into this category, as it performs zero-shot transfer by leveraging mathematical structures that are not present under linear reward decompositions. Our aim in this paper is to introduce a method that operates within the above-mentioned framework and assumptions, and to demonstrate that it outperforms the current state-of-the-art within this class of techniques.
>
> We focused our empirical comparisons on Laplacian-based methods, which have been shown in the literature—including in the FB paper itself—to be the strongest-performing techniques among SF-based zero-shot methods. As noted by the FB authors, Laplacian approaches were *“the only [ones] that perform[ed] consistently well, both over tasks and over replay buffers”* across a set of ten baselines that leverage the SF framework. Based on this, our experimental design focused on demonstrating consistent improvements over this strong and widely adopted benchmark. The corresponding results are shown in Figures 6 and 7 of our paper.
>
> For completeness, we have conducted the comparisons requested by the reviewer. Results on zero-shot performance (Q1), presented in Figure 5, show that FB—by leveraging mathematical structures not present under linear reward decompositions—sometimes, though not consistently, outperforms our method. This outcome is not altogether unexpected, as the methods operate under different modeling assumptions. Comparing our approach with FB is akin to contrasting linear value function approximation techniques (e.g., those using polynomial or Fourier bases) with function approximators operating within a different class of models, such as deep neural networks. While such comparisons can be made, they often primarily reveal differences in the assumptions and modeling tools available to each method, rather than offering a direct assessment of their effectiveness within a shared setting. Our paper aims to demonstrate consistent improvements *within* the class of SF-based zero-shot methods.
>
> That said, we do agree with the reviewer that our paper would benefit from a clearer and more thorough discussion of our experimental setup and the assumptions underlying the setting we chose to address. We will revise the manuscript accordingly, making our scope and comparison choices more explicit and adding discussion to better situate our method relative to broader approaches such as FB.

---

> ### Author Response · Authors · 2025-06-02
>
> ### Section 4 - Presentation
>
> We thank the reviewer for the helpful suggestions on how to improve Section 4. We have added a more detailed summary of all components of Algorithm 1 at the beginning of the section. Our intention with the original structure was to present the algorithm first to give readers an overview of the method, followed by detailed explanations of each component in the subsections that follow. To make this structure clearer, we now indicate, within Algorithm 1, the corresponding section where each input variable is described. We have addressed and incorporated all of the reviewer’s suggestions. We believe that Section 4 is now clearer and provides readers with a better understanding of the overall structure of the method before they move on to the details. We again thank the reviewer for these valuable suggestions, which have improved the clarity and presentation of the paper. If there are any additional parts of this section the reviewer would like to see further elaborated, please feel free to let us know: we would be happy to make further improvements.
>
> ---
>
> ### Other clarifications
>
> In addition to the points discussed above, we have also carefully addressed all remaining clarification requests raised by the reviewer:
>
> - $\gamma=1$ in Fig. 3: Thank you for pointing this out. The reviewer is correct that we used $\gamma=1$ in this figure only because the setting includes a terminal state. This choice allowed us to design a scenario in which the semantics encoded by Successor Clusters could be illustrated more intuitively. Importantly, note that our framework and theoretical results do not assume or rely on MDPs with terminal states. The goal of this figure was solely to provide readers with a clearer, more interpretable view of what Successor Clusters represent. We thank the reviewer for bringing this to our attention and for their helpful suggestion. We have updated the document accordingly.
>
> - Policy $\pi$ in Line 7 of Algorithm 1: The reviewer asked where the policy $\pi$ in Line 7 of Algorithm 1 is used. The policy $\pi$ is not used by the algorithm itself; rather, it is computed (or optimized) and returned alongside the Successor Clusters, $\psi$. In discrete-action domains, $\pi$ can be directly inferred from the SCs: $\pi(s, w) = \arg\max_{a \in \mathcal{A}} \psi(s, a; w) \cdot w$. In continuous-action domains, the actor policy $\pi(s; w) \in \arg\max_{a \in \mathcal{A}} \psi(s, a; w) \cdot w$ can be optimized using deterministic policy gradient methods, such as TD3. For more details on how $\pi$ is computed in both of these cases, please refer to Section 4.1. Following the reviewer’s suggestion and to improve clarity, we have updated Line 7 of Algorithm 1 to direct the reader to the relevant section for details on how the policy $\pi$ is computed.
>
> - On the definition of $\mathcal{W}$: Thank you for pointing out that this definition could have been made clearer. We have updated the document to improve clarity. In general, $\mathcal{W} \subset \mathbb{R}^d$. This is made more precise in Section 4.3, where we show that $\mathcal{W}$ can be restricted to the $d$-dimensional sphere of radius $\sqrt{d}$ without loss of generality. This is standard practice, e.g., in the literature on training USFAs (Touati et al., 2023). To improve clarity, we have extended the discussion of $\mathcal{W}$ by adding a more explicit explanation in the first paragraph of Section 4, before Equation 7.
>
> - How to compute Eq. (10) using $n$ samples: This is a good question. As discussed in Section 4.2, Eq. (10) is an approximation. Following the approach in Touati et al. (2023), given a target reward function $r$, we compute the weights $w$ that best linearly approximate it. While we do not have access to the exact values of $r$ everywhere—as the reviewer correctly noted—we found that using approximately $n = 10^4$ samples was sufficient in all experiments. This matches the number of samples used in the benchmark evaluation in Touati et al. (2023).
>
> - Regarding the reward functions used in the experiments (Q1): The reward functions we used are the same as those from the benchmarks conducted by Touati et al. (2023). Each domain includes four reward functions, described in detail in their work. We adopted the same set of reward functions to ensure a fair and consistent comparison. For completeness, we have included in Appendix C.2 a full list—similar to the one provided by Touati et al. (2023)—with details about the environments and the corresponding rewards. We note that these reward functions are not necessarily designed to be Lipschitz.
>
> ---
>
> We hope the comments above address the main points you raised. Please feel free to reach out with any further thoughts—we would be happy to continue this productive discussion. Thank you once again for your thoughtful feedback and for taking the time to carefully read our work and share your insights.

---

### Author Response · Authors · 2025-06-02

We thank all the reviewers for their thoughtful feedback and constructive suggestions. We have carefully and thoroughly addressed all the requested changes and clarifications, and the revised manuscript reflects these updates. Revisions are highlighted in blue for convenience.

---

### Decision · Action_Editor_x1t4 · 2025-06-10

**Recommendation:** Accept as is

**Additional Comments:**

The authors incorporated the requested changes in the revision, and therefore the paper can be accepted as is.

**Audience:**

Yes

**Audience Explanation:**

The submission presents a method for automatically constructing successor features by clustering the state-space depending on the travel distance between states and derives a bound on the linear reward approximation errors when using those features, depending on the Lipschitz constant for any Lipschitz reward function.  These results are of interest for some researchers in the field.

**Claims And Evidence:**

Yes

**Claims Explanation:**

While reviewer Mwrx spottet an error in the derivations, the author could fix it in the revision. All reviewers agree that the claims in the revision are substantiated.